# When Data Can't Meet: Estimating Correlation Across Privacy Barriers

**Abhinav Chakraborty**
Columbia University
New York, NY 10027
ac4662@columbia.edu

**Arnab Auddy**
The Ohio State University
Columbus, OH 43210
arnab.auddy@columbia.edu

**T. Tony Cai**
The Wharton School
University of Pennsylvania
Philadelphia, PA 19104
tcai@wharton.upenn.edu

## Abstract

We consider the problem of estimating the correlation of two random variables $X$ and $Y$, where the pairs $(X, Y)$ are not observed together, but are instead separated co-ordinate-wise at two servers: server 1 contains all the $X$ observations, and server 2 contains the corresponding $Y$ observations. In this vertically distributed setting, we assume that each server has its own privacy constraints, owing to which they can only share suitably privatized statistics of their own component observations. We consider differing privacy budgets $(\varepsilon_1, \delta_1)$ and $(\varepsilon_2, \delta_2)$ for the two servers and determine the minimax optimal rates for correlation estimation allowing for both non-interactive and interactive mechanisms. We also provide correlation estimators that achieve these rates and further develop inference procedures, namely, confidence intervals, for the estimated correlations. Our results are characterized by an interesting rate in terms of the sample size $n$, $\varepsilon_1$, $\varepsilon_2$, which is strictly slower than the usual central privacy estimation rates. More interestingly, we find that the interactive mechanism is always better than its non-interactive counterpart whenever the two privacy budgets are different. Results from extensive numerical experiments support our theoretical findings.

## 1 Introduction

Federated learning is a popular and extensively studied framework in modern machine learning. In traditional federated learning, due to privacy concerns, the servers are not allowed to pool raw data, but are restricted to sharing only sufficiently privatized statistics derived from the local observations. This method is particularly beneficial when training on sensitive data, such as healthcare or finance. The federated scenario is very systematically studied when the separation occurs horizontally, i.e. observations of the same set of features are binned separately into different servers. See, for example, Kairouz et al. [2021], Li et al. [2020a,b], Zhang et al. [2021] and the references therein.

To encourage collaboration on proprietary data across different organizations, however, it is often more reasonable to assume that the federation occurs "vertically", or across features. For example, in healthcare data, a hospital and a pharmaceutical company might have different pieces of information on the same patient: the hospital does not share private clinical information such as patient demographics or test results with the company, which instead has its own private information on the same patient's response to certain drugs. This new framework called vertical federated learning has recently seen studied in Chen et al. [2020], Liu et al. [2024], Wu et al. [2020], Wei et al. [2022], Yang et al. [2019], but a theoretical understanding of estimation and inference has largely been missing. This motivates the current work. We study the correlation of bivariate data from $n$ pairs of samples $(X_i, Y_i)$

39th Conference on Neural Information Processing Systems (NeurIPS 2025).

which are not *observed* together, but are instead separated into two servers as $\{X_i : 1 \leq i \leq n\}$ and $\{Y_i : 1 \leq i \leq n\}$.

To distinguish our results from the influence of estimating the marginal distributions of $X$ and $Y$, we assume that $\mathbb{E}(X) = \mathbb{E}(Y) = 0$ and $\text{Var}(X) = \text{Var}(Y) = 1$, and $(X, Y)$ are sub-Gaussian. That is, we assume that our data are pre-normalized to have mean zero and variance one. We revisit the question of normalization in the supplementary material and show both theoretically and in numerical experiments that the rate of correlation estimation is not influenced by this step. In this situation, we consider estimating $\rho = \mathbb{E}(XY)$ from the statistics shared by the two servers: viz., Server 1 releases $T_1(X_1, \ldots, X_n)$, and Server 2 releases $T_2(Y_1, \ldots, Y_n)$. To protect user privacy, we impose the differential privacy framework (see, e.g., Abowd et al. [2020], Bassily et al. [2014], Dwork [2006], Karwa and Vadhan [2017]) on $T_1$ and $T_2$; both of which must satisfy $(\varepsilon_1, \delta_1)$ and $(\varepsilon_2, \delta_2)$-DP constraints. For ease of reference, we will somewhat loosely denote the above by a server-level $(\varepsilon_1, \varepsilon_2, \delta_1, \delta_2)$-DP constraint and introduce specific definitions later. Such distributed privacy requirements are frequently used in federated learning. See, e.g., Auddy et al. [2024], Cai et al. [2024a,b,c], Shen et al. [2022], Wei et al. [2020, 2021] and references therein.

## 1.1 Main results

The key finding in this work is that the complexity of the correlation estimation in the above setup fundamentally depends on whether or not the statistics $T_1$ and $T_2$ are allowed to depend on one another. We now present our main results. Throughout this paper, we assume $\varepsilon_1, \varepsilon_2 \leq C$ for a constant $C > 0$.

### 1.1.1 Non-interactive protocol

In our first set of results, we consider estimating $\rho$ in the non-interactive (NI) framework of stricter privacy requirements, where $T_1$ and $T_2$ are constructed independently, i.e., without any interaction or information about one another. In this case, the differential privacy requirements on $T_1$ and $T_2$ are as follows. With $\mathbf{X} = (X_1, \ldots, X_n)$, $\mathbf{Y} = (Y_1, \ldots, Y_n)$, and similarly $\mathbf{X}'$, $\mathbf{Y}'$ (with one data point replaced):

$$\mathbb{P}(T_1(\mathbf{X}) \in A | \mathbf{X}) \leq \exp(\varepsilon_1)\mathbb{P}(T_1(\mathbf{X}') \in A | \mathbf{X}') + \delta_1$$
$$\mathbb{P}(T_2(\mathbf{Y}) \in A | \mathbf{Y}) \leq \exp(\varepsilon_2)\mathbb{P}(T_2(\mathbf{Y}') \in A | \mathbf{Y}') + \delta_2.$$

Let $\text{NI}(\varepsilon_1, \varepsilon_2, \delta_1, \delta_2)$ to be the class of all correlation estimators constructed using $T_1(\mathbf{X})$ and $T_2(\mathbf{Y})$ satisfying the above privacy requirement. The following theorem states the minimax rate for estimating $\rho$ in this scenario.

**Theorem 1.1.** *The minimax rate for estimating correlation $\rho$ via a non-interactive procedure satisfying server level $(\varepsilon_1, \varepsilon_2, \delta_1, \delta_2)$-DP constraints is given by*

$$\inf_{\widehat{\rho} \in \text{NI}(\varepsilon_1, \varepsilon_2, \delta_1, \delta_2)} \sup_{\rho \in [-1,1]} \mathbb{E}(\widehat{\rho} - \rho)^2 \asymp L_n\left(\frac{1}{n\varepsilon_1^2} + \frac{1}{n\varepsilon_2^2}\right)$$

*for a factor $L_n$ of order at most $O(\log(n))$, whenever $\delta_1, \delta_2 = o(n^{-1})$.*

Note that the rate does not depend on $\delta$'s. This implies that our rate matching correlation estimator achieves $(\varepsilon_1, \varepsilon_2, 0, 0)$-DP, and is still rate optimal (up to logarithmic terms) even within $\text{NI}(\varepsilon_1, \varepsilon_2, \delta_1, \delta_2)$, the class of all non-interactive estimators satisfying $(\varepsilon_1, \delta_1)$ and $(\varepsilon_2, \delta_2)$ DP constraints for $\delta_1, \delta_2$ are small positive numbers. The rate optimal estimator in this case is given by the correlation of privatized batch means from both servers.

It is useful to compare the above rate with the ones existing in the literature. Firstly, when $(X, Y)$ are jointly observed, and we impose $(\varepsilon, \delta)$-central DP constraints on $(X_i, Y_i)$, the optimal correlation estimation rate is given by $\frac{1}{n^2\varepsilon^2}$. See, e.g., Biswas et al. [2020], Cai et al. [2021]. As expected, when $\varepsilon_1 = \varepsilon_2 = \varepsilon$, this is better than the rate we observe in the current feature separated case, thus highlighting the cost of vertical federation. A second comparison can be made with component-wise local privacy rates, studied in Amorino and Gloter [2023]. The authors there show that in the vertically separated scenario, if we impose $(\varepsilon_1, 0)$ and $(\varepsilon_2, 0)$ *local* DP constraints, the minimax estimation rate for correlation is given by $\frac{1}{n\varepsilon_1^2\varepsilon_2^2}$, which is again strictly worse than the rates we find under the server level DP constraints.

### 1.1.2 Interactive protocol

We next move on to a larger class of estimators in the interactive (INT) framework, where we still require server level privacy, but one of the servers is allowed access to the privatized statistic output from the other. In other words, we allow the functions $T_1$ and $T_2$ to have one way interaction with each other. This requires making exactly one out of two possible choices. The first possibility is that when constructing $T_2$, Server 2 has access to $T_1(\mathbf{X})$, in addition to its own data $\mathbf{Y}$. The second possibility arises by analogously interchanging the roles of servers 1 and 2. To fix ideas, if we are in the first case, i.e server 2 gets to observe the transcript $T_1$, before computing $T_2$, the privacy requirements become:

$$\mathbb{P}(T_1(\mathbf{X}) \in A | \mathbf{X}) \leq \exp(\varepsilon_1)\mathbb{P}(T_1(\mathbf{X}') \in A | \mathbf{X}') + \delta_1$$

$$\mathbb{P}(T_2(\mathbf{Y}, T_1(\mathbf{X})) \in A | \mathbf{X}, \mathbf{Y}) \leq \exp(\varepsilon_2)\mathbb{P}(T_2(\mathbf{Y}', T_1(\mathbf{X})) \in A | \mathbf{X}, \mathbf{Y}') + \delta_2.$$

Replacing $\mathbf{X}$ with $\mathbf{Y}$ and the index 1 with 2 allows one to write the analogous privacy constraint in the second case where Server 1 has access to $T_2(\mathbf{Y})$. Let $\mathrm{INT}(\varepsilon_1, \varepsilon_2, \delta_1, \delta_2)$ to be the class of all correlation estimators constructed using $T_1(\mathbf{X})$ and $T_2(\mathbf{Y}, T_1(\mathbf{X}))$ satisfying the above interactive privacy requirement. The following theorem states the minimax rate for estimating $\rho$ in this scenario.

**Theorem 1.2.** *The minimax rate for estimating correlation $\rho$ via a non-interactive procedure satisfying server level $(\varepsilon_1, \varepsilon_2, \delta_1, \delta_2)$-DP constraints is given by*

$$\inf_{\widehat{\rho} \in \mathrm{INT}(\varepsilon_1, \varepsilon_2, \delta_1, \delta_2)} \sup_{\rho \in [-1,1]} \mathbb{E}\left(\widehat{\rho} - \rho\right)^2 \asymp L_n \left( \frac{1}{n(\varepsilon_1 \vee \varepsilon_2)^2} + \frac{1}{n^2 \varepsilon_1^2 \varepsilon_2^2} \right).$$

*for a factor $L_n$ of order at most $O(\log(n))$, whenever $\delta_1, \delta_2 = o(n^{-1})$.*

Note that unlike NI, in the INT rate, the dominating term depends on $\varepsilon_1 \vee \varepsilon_2$, i.e. the less stringent privacy requirement. The stronger privacy requirement i.e., $\varepsilon_1 \wedge \varepsilon_2$ appears in the second term, but its effect is mitigated by the better sample size factor $n^{-2}$. This leads to INT being a strictly better estimator than NI whenever $\varepsilon_1 \neq \varepsilon_2$. An interesting special case is when $\mathbf{X}$ are public, meaning $\varepsilon_1$ is a constant, in which case we find $(\varepsilon_2, \delta_2)$-central DP rates for correlation estimation.

The rate optimal estimator in the interactive case is borne out of a natural idea: the server with a less stringent privacy budget should share their statistics with the other server. That is, if $\varepsilon_1 > \varepsilon_2$, we should allow $T_2$ to depend on $T_1(\mathbf{X})$ and $\mathbf{Y}$. The situation is reversed if $\varepsilon_2 > \varepsilon_1$.

In addition to point estimates $\widehat{\rho}$, we also derive asymptotically valid confidence intervals in both the non-interactive (NI) and interactive (INT) scenarios. That is, we find $(\widehat{\rho}_{L,n}^{(\mathrm{NI})}, \widehat{\rho}_{U,n}^{(\mathrm{NI})})$ and $(\widehat{\rho}_{L,n}^{(\mathrm{INT})}, \widehat{\rho}_{U,n}^{(\mathrm{INT})})$ such that for fixed $\alpha \in (0, 1)$

$$\mathbb{P}\left(\widehat{\rho}_{L,n}^{(k)} \leq \rho \leq \widehat{\rho}_{U,n}^{(k)}\right) \to 1 - \alpha \quad \text{as } n \to \infty, \quad \text{for } k \in \{\mathrm{NI, INT}\}.$$

We show that our estimation methods are minimax optimal by proving corresponding lower bounds, which to the best of our knowledge, has not been established previously under central differential privacy in a vertically distributed setting. While we follow the classical Le Cam framework, our main technical contribution is a direct control of KL divergence via Fisher information curvature bounds, yielding sharp lower bounds under both non-interactive and one-way interactive protocols. These bounds match our upper bounds up to constants in the Gaussian case and up to logarithmic factors in the sub-Gaussian case. Prior works, such as Hadar et al. [2019], bound KL via mutual information in communication constraint settings; we take a more direct route tailored to central DP. Unlike local DP lower bounds in Amorino and Gloter [2023], our approach handles the more delicate structure of central privacy with vertical data splitting.

The rest of the paper is organized as follows. In Sections 2 and 3 respectively, we describe non-interactive and interactive correlation estimators for bivariate Gaussian and bivariate sub-Gaussian distributions. Section 4 provides minimax lower bounds showing that our estimation procedures are nearly optimal in all cases. Finally, Section 5 shows numerical experiments to corroborate our theoretical results. All proofs are in the supplementary material.

## 2 Non-interactive estimation methods

We first demonstrate an estimation procedure in the non-interactive paradigm. Here Server 1 and Server 2 construct and share $T_1(\mathbf{X})$ and $T_2(\mathbf{X})$ without knowledge of one another. As mentioned in the introduction $T_1(\mathbf{X})$ must satisfy $(\varepsilon_1, \delta_1)$-DP and $T_2(\mathbf{X})$ must satisfy $(\varepsilon_2, \delta_2)$-DP constraints. Our estimator is based on sharing privatized batch means. Choosing $m \geq 1$ we separate the $n$ observations in each server into batches of size $m$ as follows:

$$B_j = \{m(j-1)+1, \ldots, mj\} \quad \text{for} \quad j = 1, \ldots, k \quad \text{where } k = \lfloor \tfrac{n}{m} \rfloor. \tag{1}$$

### 2.1 Non-interactive correlation estimation for Gaussian distribution

In this subsection, we assume that $(X, Y) \sim \mathcal{N}(\mathbf{0}, \Sigma(\rho))$ with $(\Sigma(\rho))_{11} = (\Sigma(\rho))_{22} = 1$ and $(\Sigma(\rho))_{12} = \rho$, the bivariate Gaussian distribution with $\mathbb{E}(X) = \mathbb{E}(Y) = 0$, $\mathrm{Var}(X) = \mathrm{Var}(Y) = 1$ and correlation $\mathbb{E}(XY) = \rho$.

Our estimation procedure for $\rho$ is through the product of sample means across multiple batches. In order to bound the sensitivity directly, i.e., without clipping, we will use the signs of $X_i$ and $Y_i$ in place of $(X_i, Y_i)$ themselves, to compute our correlation estimator.

$$\bar{X}^{(j)} = \frac{1}{m}\sum_{i \in B_j} \mathrm{sign}(X_i), \text{ and } \bar{Y}^{(j)} = \frac{1}{m}\sum_{i \in B_j} \mathrm{sign}(Y_i) \tag{2}$$

where $B_j$ are as defined in (1) for $j = 1, \ldots, k$. To ensure $(\varepsilon_1, 0)$-DP and $(\varepsilon_2, 0)$-DP constraints each server adds Laplace noise to each batch mean and outputs the vectors $T_1(\mathbf{X}), T_2(\mathbf{Y}) \in \mathbb{R}^m$ with elements:

$$(T_1(\mathbf{X}))_j = \sqrt{m}(\bar{X}^{(j)} + Z_1^{(j)}) \quad \text{and} \quad (T_2(\mathbf{Y}))_j = \sqrt{m}(\bar{Y}^{(j)} + Z_2^{(j)}) \quad \text{for} \quad 1 \leq j \leq k,$$

where $Z_l^{(j)} \overset{indep}{\sim} \mathrm{Laplace}\left(0, \frac{2}{m\varepsilon_l}\right)$ for $l = 1, 2$. We can then compute

$$\widehat{\eta}_{XY} = \frac{1}{k}\sum_{j=1}^{k}(T_1(\mathbf{X}))_j(T_2(\mathbf{Y}))_j. \tag{3}$$

Since $(X, Y)$ are bivariate Gaussians, the covariance above satisfies

$$\mathbb{E}[\widehat{\eta}_{XY}] = 2\mathbb{P}(XY > 0) - 1 = 1 - \frac{2\arccos(\rho)}{\pi}, \tag{4}$$

which leads to the method-of-moments based private correlation estimator:

$$\widehat{\rho}_{\mathrm{NI}}^{(G)} := \cos\left(\frac{\pi}{2}(1 - \widehat{\eta}_{XY}^{(P)})\right) = \sin\left(\frac{\pi\widehat{\eta}_{XY}^{(P)}}{2}\right).$$

We would like to emphasize that (4) is precisely where we use the assumption of Gaussianity on $(X, Y)$. Since the bivariate distribution is completely known once $\rho$ is specified, we can explicitly write $\mathbb{P}(XY > 0)$ as a function of $\rho$, which in turn enables our sign-based estimation procedure. While this can be extended to other bivariate families which are specified by a single correlation parameter $\rho$, we do not discuss these details for brevity.

To create confidence intervals for $\rho$, let us define $S_\eta^2$ to be the sample variance of $\{(T_1(\mathbf{X}))_j(T_2(\mathbf{Y}))_j : 1 \leq j \leq k\}$. Then we can define the confidence interval:

$$\mathrm{CI}_{\mathrm{NI}}^{(G)}(\alpha) := \left(\widehat{\rho}_{\mathrm{NI}}^{(G)} - \frac{\pi S_\eta \sqrt{1-(\widehat{\rho}_{\mathrm{NI}}^{(G)})^2}}{2\sqrt{k}} z_{1-\alpha/2}, \widehat{\rho}_{\mathrm{NI}}^{(G)} + \frac{\pi S_\eta \sqrt{1-(\widehat{\rho}_{\mathrm{NI}}^{(G)})^2}}{2\sqrt{k}} z_{1-\alpha/2}\right) \tag{5}$$

where $z_{1-\alpha/2}$ is the $(1 - \alpha/2)$-th quantile of the standard Normal distribution.

### 2.2 Non-interactive correlation estimation for sub-Gaussian distributions

In general, we would deal with non-Gaussian data, and thus the sign-based procedure of the previous section would not be exact anymore. We will use a clipping based estimator for this case. For clipping parameters $\lambda_1, \lambda_2 > 0$ to be chosen later we replace (2) by

$$\bar{X}^{(j)} = \frac{1}{m}\sum_{i \in B_j} \mathrm{sign}(X_i)(|X_i| \wedge \lambda_1) \text{ and } \bar{Y}^{(j)} = \frac{1}{m}\sum_{i \in B_j} \mathrm{sign}(Y_i)(|Y_i| \wedge \lambda_2) \tag{6}$$

where $B_j$ are as defined in (1) for $j = 1, \ldots, k$. As before, each server adds Laplace noise to each batch mean and shares:

$$(T_1(\mathbf{X}))_j = \sqrt{m}(\bar{X}^{(j)} + Z_1^{(j)}) \quad \text{and} \quad (T_2(\mathbf{Y}))_j = \sqrt{m}(\bar{Y}^{(j)} + Z_2^{(j)}) \quad \text{for} \quad 1 \le j \le k,$$

where $Z_l^{(j)} \overset{indep}{\sim} \text{Laplace}\left(0, \frac{2\lambda_l}{m\varepsilon_l}\right)$ for $l = 1, 2$. Then we will estimate $\rho$ by the quantity:

$$\widehat{\rho}_{\text{NI}}^{(SG)} = \frac{1}{k} \sum_{j=1}^{k} (T_1(\mathbf{X}))_j (T_2(\mathbf{Y}))_j. \tag{7}$$

Once again defining $S_\rho^2$ to be the sample variance of $\{(T_1(\mathbf{X}))_j (T_2(\mathbf{Y}))_j : 1 \le j \le k\}$, we have the confidence interval:

$$\text{CI}_{\text{NI}}^{(SG)}(\alpha) := \left(\widehat{\rho}_{\text{NI}}^{(SG)} - \frac{S_\rho}{\sqrt{k}} z_{1-\alpha/2}, \widehat{\rho}_{\text{NI}}^{(SG)} + \frac{S_\rho}{\sqrt{k}} z_{1-\alpha/2}\right) \tag{8}$$

where $z_{1-\alpha/2}$ is the $(1-\alpha/2)$-th quantile of the standard Normal distribution. The following theorem states the results for correlation estimator under non-interactive protocol.

**Theorem 2.1.** *The following results hold on the estimation error of $\rho$ using a non-interactive componentwise privacy constrained estimator.*

1. *When $(X, Y) \sim \mathcal{N}(\mathbf{0}, \Sigma(\rho))$ with $(\Sigma(\rho))_{11} = (\Sigma(\rho))_{22} = 1$ and $(\Sigma(\rho))_{12} = \rho$, the estimator $\widehat{\rho}_{\text{NI}}^{(G)}$ described in Section 2.1 satisfies $\widehat{\rho}_{\text{NI}}^{(G)} \in \text{NI}(\varepsilon_1, \varepsilon_2, \delta_1, \delta_2)$ and*

$$\mathbb{E}(\widehat{\rho}_{\text{NI}}^{(G)} - \rho)^2 \lesssim \frac{1}{n}\left(\frac{1}{\varepsilon_1^2} + \frac{1}{\varepsilon_2^2}\right) \quad \text{if} \quad m = \left\lfloor \frac{8}{\varepsilon_1 \varepsilon_2} \right\rfloor \vee 1.$$

2. *When $(X, Y)$ have mean zero, variance one, $X$ is $\eta_1$-sub-Gaussian, $Y$ is $\eta_2$-sub-Gaussian, and $\mathbb{E}[XY] = \rho$, the estimator $\widehat{\rho}_{\text{NI}}^{(SG)}$ described in Section 2.2 satisfies $\widehat{\rho}_{\text{NI}}^{(SG)} \in \text{NI}(\varepsilon_1, \varepsilon_2, \delta_1, \delta_2)$ and*

$$\mathbb{E}(\widehat{\rho}_{\text{NI}}^{(SG)} - \rho)^2 \lesssim \frac{\log(n)}{n}\left(\frac{1}{\varepsilon_1^2} + \frac{1}{\varepsilon_2^2}\right) \quad \text{if} \quad m = \left\lfloor \frac{\lambda_1 \lambda_2}{\varepsilon_1 \varepsilon_2} \right\rfloor \vee 1,$$

*$\lambda_1 = 2\eta_1 \sqrt{\log(n)}$, and $\lambda_2 = 2\eta_2 \sqrt{\log(n)}$.*

3. *For any fixed $\alpha \in (0, 1)$, the confidence intervals defined in (5) and (8) satisfy $\mathbb{P}(\rho \in \text{CI}_{\text{NI}}^{(k)}(\alpha)) \to 1 - \alpha$ as $n \to \infty$, for $k \in \{G, SG\}$.*

## 3 Interactive estimation methods

We now show that the rates in the previous section can be improved if we allow a one-step interactive scheme between the two servers. To fix ideas, suppose $\varepsilon_1 > \varepsilon_2$, i.e., the privacy requirement in the first server are less stringent than that in the second one. We will then share the private transcripts involving $X$ to the second server containing the $Y$ observations. This leads to an estimation error rate that improves over the non-interactive protocol.

### 3.1 Interactive correlation estimation for Gaussian distribution

In this case, our interactive estimator based on signs of $(X, Y)$ is as follows. Server 1 first communicates to Server 2 the privatized sign vector $T_1(\mathbf{X})$ with elements:

$$(T_1(\mathbf{X}))_i = \frac{\exp(\varepsilon_1) + 1}{(\exp(\varepsilon_1) - 1)}(2S_i - 1)\operatorname{sign}(X_i) \quad \text{for} \quad i = 1, \ldots, n$$

where $S_i \overset{iid}{\sim} \text{Bernoulli}\left(\frac{\exp(\varepsilon_1)}{\exp(\varepsilon_1)+1}\right)$ are independent sign flips introduced by Server 1 to protect the privacy of $X_i$. Given $T_1(\mathbf{X})$ the second server first computes the covariance

$$\widehat{\eta}_{XY,\text{int}} = \frac{1}{n} \sum_{i=1}^{n} (T_1(\mathbf{X}))_i \operatorname{sign}(Y_i)$$

and then outputs the privatized version

$$T_2(\mathbf{Y}, T_1(\mathbf{X})) := \widehat{\eta}_{XY,\text{int}} + Z \quad \text{where } Z \sim \text{Laplace}\left(0, \frac{2(\exp(\varepsilon_1)+1)}{n(\exp(\varepsilon_1)-1)\varepsilon_2}\right). \tag{9}$$

As before we then have the private correlation estimator

$$\widehat{\rho}_{\text{INT}}^{(G)} = \sin\left(\frac{\pi \widehat{\eta}_{XY,\text{int}}^{(P)}}{2}\right).$$

Similar to the non-interactive case, defining $\widehat{\sigma}_\eta^2 := 1 - \left(\frac{\exp(\varepsilon_1)-1}{\exp(\varepsilon_1)+1}\right)^2 (\widehat{\eta}_{XY,\text{int}}^{(P)})^2$ allows the confidence interval given by the following.

1. If $c_* = \lim_{n\to\infty} \frac{2}{\sqrt{n}\sigma_\eta \varepsilon_2}$ is finite, then the CI is

$$\left(\widehat{\rho}_{\text{INT}}^{(G)} \mp \frac{\pi \widehat{\sigma}_\eta \sqrt{1-(\widehat{\rho}_{\text{INT}}^{(G)})^2}}{2\sqrt{n}} \left(\frac{\exp(\varepsilon_1)+1}{\exp(\varepsilon_1)-1}\right) F_*^{-1}(1-\alpha/2)\right) \tag{10}$$

   where for any $x \in \mathbb{R}$ we define $F_*(x) := \mathbb{P}(Z_{XY} + \widehat{c}_* Z_{\text{Lap}} \le x)$ for $\widehat{c}_* = 2/(\sqrt{n}\widehat{\sigma}_\eta \varepsilon_2)$ and $Z_{\text{Lap}} \sim \text{Laplace}(0,1)$.

2. If $\frac{1}{\sqrt{n}\varepsilon_2}$ diverges as $n \to \infty$, then the CI is

$$\left(\widehat{\rho}_{\text{INT}}^{(G)} \pm \frac{\pi \sqrt{1-(\widehat{\rho}_{\text{INT}}^{(G)})^2}}{n\varepsilon_2} \left(\frac{\exp(\varepsilon_1)+1}{\exp(\varepsilon_1)-1}\right) \log(\alpha)\right). \tag{11}$$

### 3.2 Interactive correlation estimation for sub-Gaussian distributions

Following previous sections, Server 1 will send to Server 2 the vector of privatized clipped observations $T_1(\mathbf{X}) \in \mathbb{R}^n$ with elements $(T_1(\mathbf{X}))_i = [X_i]_{\lambda_1} + Z_{1i}$ for a clipping parameter $\lambda_1 > 0$ and $Z_{1i} \overset{iid}{\sim} \text{Laplace}(2\lambda_1/\varepsilon_1)$ for $i = 1, \ldots, n$. Then Server 2 can construct

$$\widehat{\rho}_{\text{INT}}^{(SG)} = \frac{1}{n} \sum_{i=1}^{n} [(T_1(\mathbf{X}))_i Y_i]_{\lambda_2} + Z_2.$$

In the above $[x]_t := \text{sign}(x)(|x| \wedge t)$ for any $x \in \mathbb{R}$ and $t > 0$. Here $Z_2 \sim \text{Laplace}(2\lambda_2/n\varepsilon_2)$ is Laplace noise added to ensure DP requirements. In addition to $\widehat{\rho}_{\text{INT}}^{(SG)}$, Server 2 also outputs a privatized sample variance $S_\rho^2$ of $[(T_1(\mathbf{X}))_i Y_i]_{\lambda_2}$ for $i = 1, \ldots, n$. Then we have the confidence interval constructed as follows:

1. If $c_* = \lim_{n\to\infty} \frac{2\lambda_2}{\sqrt{n}\sigma_\rho \varepsilon_2}$ is finite, then the CI is

$$\left(\widehat{\rho}_{\text{INT}}^{(SG)} - \frac{S_\rho}{\sqrt{n}} F_*^{-1}(1-\alpha/2), \widehat{\rho}_{\text{int}}^{(SG)} + \frac{S_\rho}{\sqrt{n}} F_*^{-1}(1-\alpha/2)\right) \tag{12}$$

   where for any $x \in \mathbb{R}$ we define $F_*(x) := \mathbb{P}(Z_{XY} + \widehat{c}_* Z_{\text{Lap}} \le x)$ for $\widehat{c}_* = 2\lambda_2/(\sqrt{n}S_\rho \varepsilon_2)$, and $Z_{\text{Lap}} \sim \text{Laplace}(0,1)$.

2. If $\frac{\lambda_2}{\sqrt{n}\varepsilon_2}$ diverges as $n \to \infty$, then the CI is

$$\left(\widehat{\rho}_{\text{INT}}^{(SG)} + \frac{\lambda_2}{n\varepsilon_2} \log(\alpha), \widehat{\rho}_{\text{int}}^{(SG)} - \frac{\lambda_2}{n\varepsilon_2} \log(\alpha)\right). \tag{13}$$

The following theorem states the results for correlation estimator under the interactive protocol.

**Theorem 3.1.** *The following results hold on the estimation error of $\rho$ using the above privacy constrained interactive estimator.*

1. *When $(X,Y) \sim \mathcal{N}(\mathbf{0}, \Sigma(\rho))$ with $(\Sigma(\rho))_{11} = (\Sigma(\rho))_{22} = 1$ and $(\Sigma(\rho))_{12} = \rho$, the estimator $\widehat{\rho}_{\text{INT}}^{(G)}$ described in Section 3.1 satisfies $\widehat{\rho}_{\text{INT}}^{(G)} \in \text{INT}(\varepsilon_1, \varepsilon_2, \delta_1, \delta_2)$ and*

$$\mathbb{E}(\widehat{\rho}_{\text{INT}}^{(G)} - \rho)^2 \lesssim \frac{1}{n(\varepsilon_1 \vee \varepsilon_2)^2} + \frac{1}{n^2 \varepsilon_1^2 \varepsilon_2^2}.$$

2. *When $(X, Y)$ have mean zero, variance one, $X$ is $\eta_1$-sub-Gaussian, $Y$ is $\eta_2$-sub-Gaussian, and $\mathbb{E}[XY] = \rho$, the estimator $\widehat{\rho}_{\mathrm{INT}}^{(SG)}$ described in Section 3.2 satisfies $\widehat{\rho}_{\mathrm{INT}}^{(SG)} \in \mathrm{INT}(\varepsilon_1, \varepsilon_2, \delta_1, \delta_2)$ and*

$$\mathbb{E}(\widehat{\rho}_{\mathrm{INT}}^{(SG)} - \rho)^2 \lesssim \frac{1}{n(\varepsilon_1 \vee \varepsilon_2)^2} + \frac{1}{n^2 \varepsilon_1^2 \varepsilon_2^2}$$

*if $\lambda_1 = 2\eta_1 \sqrt{\log(n)}$ and $\lambda_2 = 4(\eta_2 \vee 1)(\log(n))^2/(\varepsilon_1 \wedge 1)$.*

3. *For any fixed $\alpha \in (0, 1)$, under their respective assumptions, the confidence intervals defined in (10), (11), (12), and (13) satisfy $\mathbb{P}(\rho \in \mathrm{CI}_{\mathrm{INT}}^{(k)}(\alpha)) \to 1 - \alpha$ as $n \to \infty$, for $k \in \{G, SG\}$.*

# 4 Minimax lower bounds

In this section, we show that the private correlation estimators derived in the previous section are in fact minimax optimal in many cases. Our proof strategy is based on bounding Fisher information of the private transcripts and then using Van Trees inequality. We recall some standard results from parameter estimation theory in the next subsection.

## 4.1 Fisher information and Van Trees inequality

Let $\theta$ be a real-valued parameter taking an unknown value in some interval $[a, b]$. We observe some random variable (or vector) $X$ with distribution $P(x|\theta)$ parameterized by $\theta$.

Assume that $P(\cdot|\theta)$ is absolutely continuous with respect to a reference measure $\mu$, for each $\theta \in [a, b]$, and $\frac{dP(\cdot|\theta)}{d\mu}(x)$ is differentiable with respect to $\theta \in (a, b)$ for $\mu$-almost all $x$. Then the *Fisher information* of $\theta$ w.r.t. $X$, denoted as $I_F(X; \theta)$, is defined as

$$I_F(X; \theta) \triangleq \int \left( \frac{\partial}{\partial \theta} \ln \frac{dP(\cdot|\theta)}{d\mu}(x) \right)^2 dP(x|\theta). \tag{14}$$

The following inequality is well-known. See for example Gill and Levit [1995].

**Lemma 1** (Van Trees inequality)**.** *Let $\theta$ be a real parameter with prior density $\zeta$ supported on $[a, b] \subset \mathbb{R}$, and let $X \sim P(\cdot \mid \theta)$ with conditional density $p(x \mid \theta) = \frac{dP(\cdot|\theta)}{d\mu}(x)$. Under some regularity conditions we have that for every estimator $\widehat{\theta} = \widehat{\theta}(X)$ with $\mathbb{E}[(\widehat{\theta} - \theta)^2] < \infty$ under the joint law of $(X, \theta)$ satisfies*

$$\mathbb{E}\left[(\widehat{\theta} - \theta)^2\right] \geq \frac{1}{\mathbb{E}_\theta[I_F(X; \theta)] + I_F(\zeta)}, \qquad \mathbb{E}_\theta[I_F(X; \theta)] = \int_a^b I_F(X; \theta)\, \zeta(\theta)\, d\theta, \tag{15}$$

*where $I_F(\zeta) := \int_a^b \frac{\left(\zeta'(\theta)\right)^2}{\zeta(\theta)} d\theta$ is the prior Fisher information.*

The "regularity conditions" in Lemma 1 are to ensure that one can apply the dominated convergence theorem to exchange certain integrals and differentiations in the calculus. See for example Vaart [1998]. Additionally, assume that

$$I_F(X; \theta + \epsilon) = I_F(X; \theta)(1 + \eta(\epsilon)) \tag{16}$$

where $\eta(\epsilon) < C_\eta$ for all $|\epsilon| < c_0$ for some numerical constants $c_0 < 1$ and $C_\eta > 0$.

## 4.2 Non interactive

For the non-interactive protocols the servers output transcripts $T_1$ and $T_2$ which are $(\varepsilon_1, \delta_1)$ and $(\varepsilon_2, \delta_2)$-DP respectively. The transcripts are based on;y on the data from their own servers. An estimator $\widehat{\rho}$ is then calculated after combining $T_1$ and $T_2$.

Our lower bound is shown by the difficulty of correlation estimation when $\rho = 0$. Let us denote the transcripts by $T \equiv (T_1, T_2)$. As a first step, the next lemma shows that $I_F(T; 0)$ is smaller than a quantity involving the sample size $n$ and the privacy parameters $\varepsilon_1, \varepsilon_2$.

**Lemma 2.** *Assume that for $k = 1, 2$ , $\delta_k \log(1/\delta_k) = O(\varepsilon_k^2)$. Let us denote the Fisher information for the transcripts $T$ by $I_F(T; \rho)$. We have that*

$$I_F(T; 0) \leq \frac{8}{\pi}(n\varepsilon_1^2 \wedge n\varepsilon_2^2).$$

The local regularity assumption in (16) at $\rho = 0$ ensures that up to a constant factor, the bound from the above lemma carries over to $I_F(T; \rho)$ for $|\rho| \leq c_0$. For a suitable choice of prior density $\zeta$ this in turn implies an upper bound on $\mathbb{E}_0[I_F(T; 0)]$ and allows us to complete the proof by using Van-Trees inequality (Lemma 1). We then have the following lower bound on the minimax risk for estimating $\rho$ in the non-interactive setting.

**Theorem 4.1.** *Assume that $\delta_k = o(n^{-1-\omega})$ for $k = 1, 2$ and $n(\varepsilon_1^2 \wedge \varepsilon_2^2) \to \infty$. Then for non interactive protocols the minimax rate is lower bounded by*

$$\inf_{\widehat{\rho} \in \mathrm{NI}(\varepsilon_1, \varepsilon_2, \delta_1, \delta_2)} \sup_{\rho \in [-1,1]} |\widehat{\rho} - \rho|^2 \gtrsim \frac{1}{n} + \frac{1}{n\varepsilon_1^2} + \frac{1}{n\varepsilon_2^2}.$$

**Remark 4.1.** *The assumption $n(\varepsilon_1^2 \wedge \varepsilon_2^2) \to \infty$ assumes that the minimax rate is going to zero ensuring consistent estimation of $\rho$ in the first place.*

### 4.3 Interactive

We next allow one way interaction among the servers where either of the server can share its transcripts with the other server. Let us denote the set of protocols which allow allow interaction from server 1 to 2 as $\Pi_{1 \to 2}$, i.e server 2 gets to observe the transcript $T_1$, before computing $T_2$. We first show the following upper bound on $I_F(\Pi_{1 \to 2}; 0)$.

**Lemma 3.** *Assume that $\delta_1 \log(1/\delta_1) = o(\varepsilon_1^2)$ and $\delta_2 \log(1/\delta_2)^2 = o(n\varepsilon_1^2 \varepsilon_2^2)$. Let us denote the Fisher information for the transcripts $\Pi_{1 \to 2}$ by $I_F(\Pi_{1 \to 2}; \rho)$. We have that*

$$I_F(\Pi_{1 \to 2}; 0) \leq n\varepsilon_1^2 \wedge n^2 \varepsilon_1^2 \varepsilon_2^2.$$

If we denote the protocol which allow interaction from server 2 to 1 we can show that $I_F(\Pi_{2 \to 1}; 0) \leq n\varepsilon_2^2 \wedge n^2 \varepsilon_1^2 \varepsilon_2^2$. Since we allow for either of the protocols $\Pi \equiv (\Pi_{1 \to 2}, \Pi_{2 \to 1})$ we have that

$$I_F(\Pi; 0) \leq I_F(\Pi_{1 \to 2}; 0) \vee I_F(\Pi_{2 \to 1}; 0) \leq (n\varepsilon_1^2 \wedge n^2 \varepsilon_1^2 \varepsilon_2^2) \vee (n\varepsilon_2^2 \wedge n^2 \varepsilon_1^2 \varepsilon_2^2). \tag{17}$$

Similar to the non-interactive case we can then use the local regularity assumption in (16) and a suitable prior density $\zeta$ with Van Trees inequality, leading to the following lower bound on the minimax risk in the interactive setting.

**Theorem 4.2.** *Assume that for $k = 1, 2$ $\delta_k = o(n^{-1-\omega})$ for $\omega > 0$, $n(\varepsilon_1^2 \vee \varepsilon_2^2) \to \infty$ and $n^2 \varepsilon_1^2 \varepsilon_2^2 \to \infty$. Then for interactive protocols the minimax rate is lower bounded by*

$$\inf_{\widehat{\rho} \in \mathrm{INT}(\varepsilon_1, \varepsilon_2, \delta_1, \delta_2)} \sup_{\rho \in [-1,1]} |\widehat{\rho} - \rho|^2 \gtrsim \frac{1}{n} + \frac{1}{n(\varepsilon_1^2 \vee \varepsilon_2^2)} + \frac{1}{n^2 \varepsilon_1^2 \varepsilon_2^2}.$$

**Remark 4.2.** *The assumption $n(\varepsilon_1^2 \vee \varepsilon_2^2) \to \infty$ and $n^2 \varepsilon_1^2 \varepsilon_2^2 \to \infty$ assumes that the minimax rate is going to zero, ensuring consistent estimation of $\rho$ in the first place.*

## 5 Numerical experiments

We evaluate our **non–interactive sign–batch** (NI) and **interactive sign–flip** (INT) estimators across different parameter settings. All our codes can be found at https://github.com/abhinavc3/distributed-correlation.

### 5.1 Simulation experiments

In our experiments we write non-normalized to mean that the mean and variances of the marginal distributions are known, and normalized to mean that they are unknown and estimated. We use two generative models.

- **Gaussian:** $(X, Y) \sim \mathcal{N}(\mu, 2\Sigma(\rho))$ with $\mu = (0.5, 0.5)^\top$, and $\Sigma(\rho)$ given by $\mathrm{Var}(X) = \mathrm{Var}(Y) = 1$ and $\mathrm{Corr}(X, Y) = \rho$. We run each estimator *with* and *without* the private normalization step.

- **Bounded-factor (sub-Gaussian):** $X = U + E_1$, $Y = U + E_2$ with $U \sim \mathrm{Unif}[-\sqrt{3\rho}, \sqrt{3\rho}]$ and $E_i \sim \mathrm{Unif}[-\sqrt{3(1-\rho)}, \sqrt{3(1-\rho)}]$, so each marginal is centred, variance–one, and bounded hence sub–Gaussian.

For every design point we record mean–squared error (MSE), average confidence–interval (CI) length, empirical coverage $(1 - \alpha = 0.95)$ and the mean CI offset band $\mathbb{E}[\mathrm{CI}_L - \rho] \to \mathbb{E}[\mathrm{CI}_U - \rho]$ where $\mathrm{CI}_L$ and $\mathrm{CI}_U$ are the upper and lower confidence bars. In practice it is sufficient to use the confidence intervals from (10) and (12) since (11) and (13) are respectively the limiting versions of the above two.

**Parameter Grid.** We vary our parameters as below, with 250 replications for each cell:

- Sample size: $n \in \{1000, 1500, 2500, 4000, 6000, 9000\}$.
- Correlation: $\rho \in \{0, 0.15, 0.3, 0.4, 0.5, 0.65, 0.8, 0.9\}$.
- Privacy budget: $(\varepsilon_1, \varepsilon_2) \in \{(0.5, 0.5), (1, 1), (1.5, 0.5)\}$.

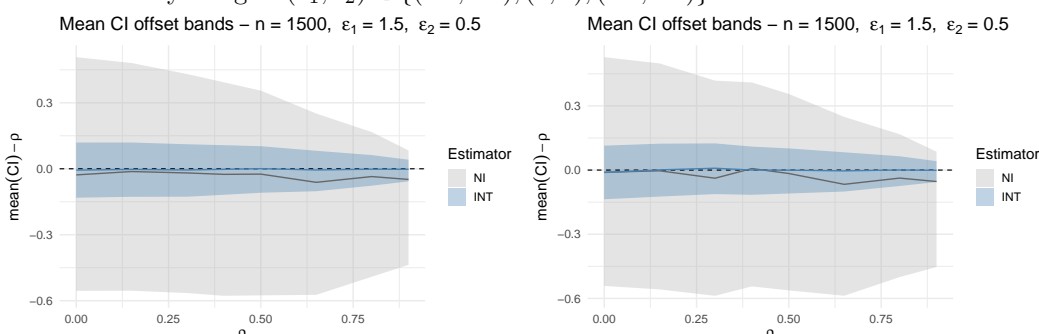

Figure 1: **Gaussian,** $n = 1500$, $(\varepsilon_1, \varepsilon_2) = (1.5, 0.5)$. Mean CI–offset bands for NI (grey) and INT (blue). Left: *without* normalization. Right: *with* private normalization. Curves overlap.

Figure 1 compares the mean CI offset bands for $n = 1500$ and the budget $(\varepsilon_1, \varepsilon_2) = (1.5, 0.5)$. With and without normalization the ribbons coincide, indicating that *private normalization is cost–free.* Figure 2 shows CI width and coverage versus $n$ at $\rho = 0.5$; both variants adhere to the nominal 95% band. Figure 3 confirms that INT is uniformly more efficient than NI, while normalization leaves MSE unchanged (largest relative difference $< 2\%$).

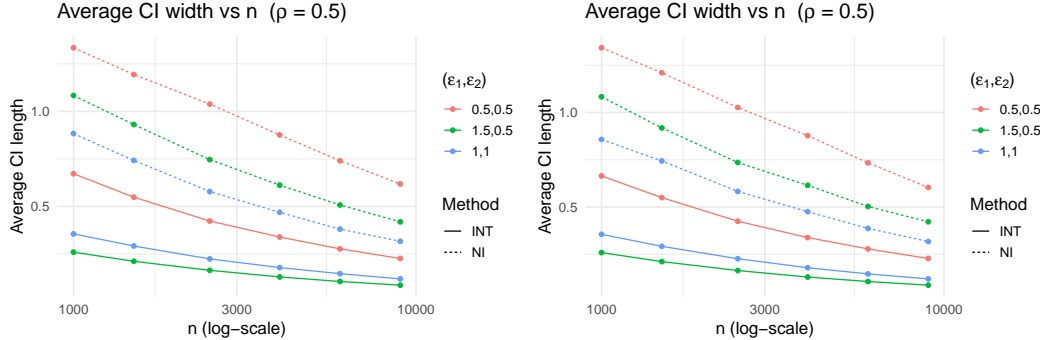

Figure 2: **Gaussian,** $\rho = 0.5$. Average CI length versus $n$. Left: *no* normalization; right: *with* normalization. Normalization has no discernible effect; INT yields shorter CIs. The coverage probabilities are above 0.91 for all CIs.

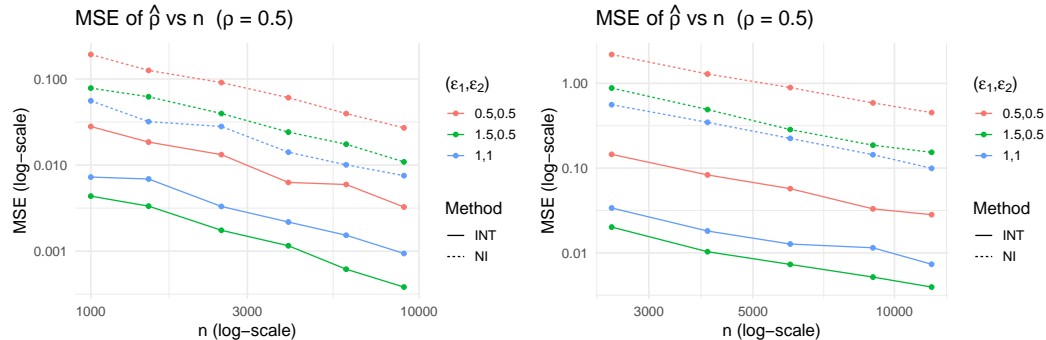

Figure 3: **Gaussian** (left) and **Bounded Factor** (right) MSE, $\rho = 0.5$. MSE versus $n$ (log–log). INT dominates NI
We repeat the study with the bounded-factor DGP. The qualitative picture is the same: INT enjoys narrower CIs and lower MSE , and both estimators achieve nominal coverage.

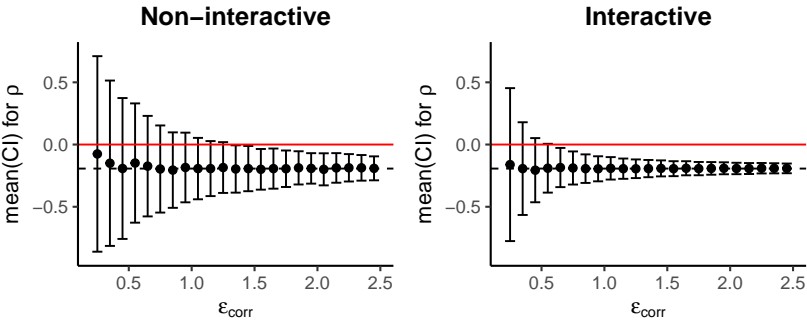

Figure 4: Mean confidence interval bands for non-interactive (left) and interactive (right) methods for estimating the correlation between age and BMI in the *Health and Retirement Study (HRS)* data. The black dotted line indicates the non-private estimator.

For the sake of brevity we only show the MSE plots (Figure 3 right). The CI bands, coverage and width plots are deferred to the supplementary material.

## 5.2 Real data experiments

We illustrate our methods using data from the *Health and Retirement Study (HRS)*, a longitudinal survey of older adults in the United States. We focus on two variables—age and body mass index (BMI)—from Wave 2 (year 1993-94) corresponding to around 20k individuals. In this demographic, age and BMI are known to exhibit a mild negative correlation.

We consider a distributed scenario in which the two variables reside on separate servers, and the goal is to estimate their Pearson correlation coefficient $\rho$. Each server first applies a Central differentially private (CDP) normalization so that the privatized features have approximately zero mean and unit variance. Specifically, we allocate $\varepsilon = 0.1$ for each of the mean and standard deviation estimates. The clipping bounds are chosen based on domain knowledge—[45, 90] for age and [15, 35] for BMI—demonstrating a setting where the privacy mechanism leverages prior information rather than data-dependent thresholds.

After normalization, we apply both the non-interactive (NI) and interactive (INT) protocols to obtain private confidence intervals for the estimated correlation $\widehat{\rho}$. We compare these to the non-private benchmark while varying the privacy budget $\varepsilon_{\text{corr}}$, keeping it equal across the two servers. Results are given in Figure 4. As $\varepsilon_{\text{corr}}$ increases, the private intervals contract and concentrate around the non-private $\rho$. Moreover, for a fixed $\varepsilon_{\text{corr}}$, the INT intervals are consistently shorter than their NI counterparts. Notably, at $\varepsilon_{\text{corr}} = 1$, the interactive CI excludes zero while the non–interactive CI includes it—illustrating that privacy noise can increase uncertainty and, in some cases, prevent rejection of the null hypothesis $\rho = 0$.

## 6 Discussion

Across both distributions and all privacy budgets explored, INT consistently outperforms NI, while the required private normalization step incurs *no measurable loss* in bias, MSE or interval width. These findings support the theoretical claim that normalization's privacy cost is dominated by the subsequent correlation release.

We discuss two important directions of future work. First, allowing multiple features per server—rather than a single feature—introduces new challenges, particularly in handling inter-feature correlations and maintaining privacy in higher dimensions. Second, extending our methods to heavy-tailed distributions would broaden applicability, as such data often arise in practice and require more robust estimation techniques.

## Acknowledgements

The research was supported in part by NSF grant NSF DMS-2413106 and NIH grants R01-GM123056 and R01-GM129781.

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

# A Implementation details

Since the mean and variance of each server can be computed under the central differential privacy (CDP) framework, we adopt estimators similar to those proposed in Karwa and Vadhan [2017] for our simulation study. After obtaining these estimators, we standardize the data and use the resulting values for downstream analysis.

Additionally, to improve the stability of our estimators, we incorporate intermediate clipping steps in our simulation study. For example, in the Gaussian case, we clip the mean of the signs to the interval $[-1, 1]$ before applying the *sin* transformation. In the sub-Gaussian case, we clip the final estimator to $[-1, 1]$.

## A.1 Additional simulation study

Here we collect the additional plots and results pertaining to our simulation study.

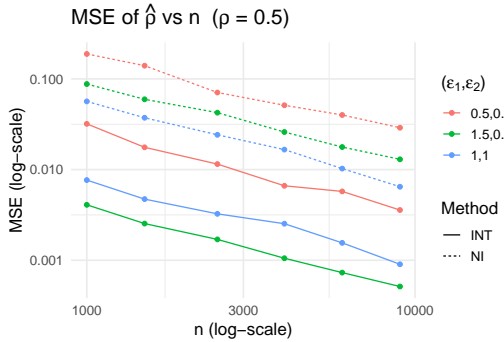

Figure 5: **Gaussian Normalized MSE,** $\rho = 0.5$**.** MSE versus $n$ (log–log).

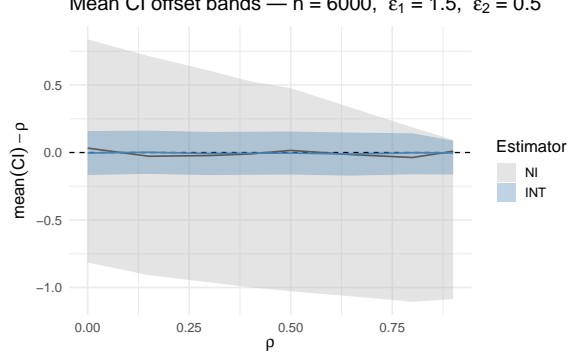

Figure 6: **Bounded-factor,** $n = 6000$, $(\varepsilon_1, \varepsilon_2) = (1.5, 0.5)$**.** Mean CI offset bands for NI and INT.

# B Proofs

*Proof of Theorem 1.1.* The proof of this theorem follows from parts 1 and 2 of Theorem 2.1 and Theorem 4.1. □

*Proof of Theorem 1.2.* The proof of this theorem follows from parts 1 and 2 of Theorem 3.1 and Theorem 4.2. □

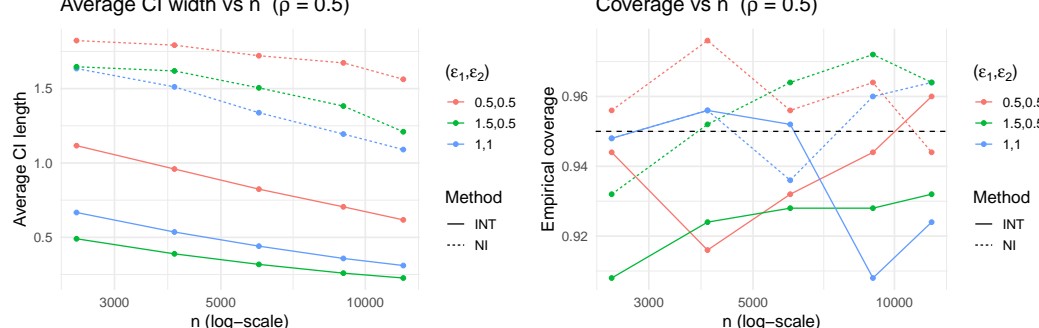

Figure 7: **Bounded-factor,** $\rho = 0.5$. CI length (left) and coverage (right) versus $n$.

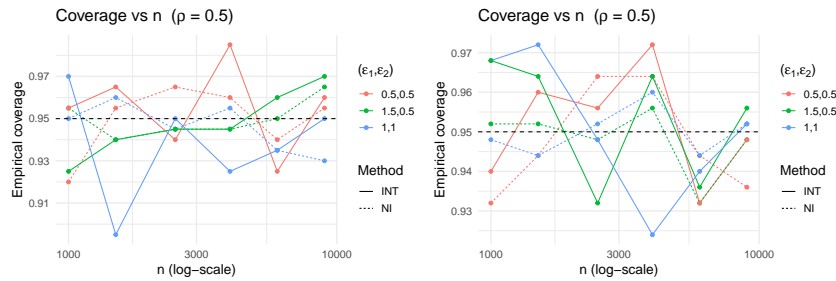

Figure 8: **Gaussian,** $\rho = 0.5$. Empirical coverage versus $n$. Left: *no* normalisation; right: *with* normalisation. Normalisation has no discernible effect; INT yields shorter CIs. The coverage probabilities are above 0.91 for all CIs.

### B.1 Proofs of upper bound results

*Proof of Theorem 2.1.* We prove the two statements in the theorem separately.

1. It is straightforward to check that

$$\mathbb{E}[\widehat{\eta}_{XY}] = m\mathbb{E}[\bar{X}^{(1)}(\bar{Y}^{(1)})] = m \cdot \frac{1}{m^2} \sum_{i=1}^{m} \mathbb{E}[\operatorname{sign}(X_i)\operatorname{sign}(Y_i)]$$

$$= 2\mathbb{P}(XY > 0) - 1 = \frac{2\arccos(-\rho)}{\pi} - 1 = 1 - \frac{2\arccos(\rho)}{\pi}.$$

To bound the error in estimating $\rho$ by $\widehat{\rho}^{(P)}$ we therefore note that

$$\left|\widehat{\rho}_{\mathrm{NI}}^{(G)} - \rho\right| = \left|\sin\left(\frac{\pi\widehat{\eta}_{XY}}{2}\right) - \sin\left(\frac{\pi\mathbb{E}[\widehat{\eta}_{XY}]}{2}\right)\right|$$

$$= \frac{\cos(\xi) \cdot \pi|\widehat{\eta}_{XY} - \mathbb{E}[\widehat{\eta}_{XY}]|}{2} \leq \frac{\pi}{2}|\widehat{\eta}_{XY}^{(P)} - \mathbb{E}[\widehat{\eta}_{XY}]|. \qquad (18)$$

where $\xi = t\widehat{\eta}_{XY} + (1-t)\mathbb{E}[\widehat{\eta}_{XY}]$ for some $t \in [0,1]$. Thus,

$$\mathbb{E}|\widehat{\rho}_{\mathrm{NI}}^{(G)} - \rho|^2 \leq \frac{\pi^2}{4}\mathbb{E}|\widehat{\eta}_{XY} - \mathbb{E}[\widehat{\eta}_{XY}]|^2 = \frac{\pi^2}{4}\operatorname{Var}(\widehat{\eta}_{XY} - \mathbb{E}[\widehat{\eta}_{XY}])$$

$$\leq \frac{\pi^2 m^2}{4k}\left[\mathbb{E}(\bar{X}^{(1)} + Z_1^{(1)})^4\right]^{1/2}\left[\mathbb{E}(\bar{Y}^{(1)} + Z_2^{(1)})^4\right]^{1/2}$$

$$\leq \frac{\pi^2 m^2}{4k}\left(\frac{3}{m} + \frac{8}{m^2\varepsilon_1^2}\right)\left(\frac{3}{m} + \frac{8}{m^2\varepsilon_2^2}\right)$$

$$= \frac{\pi^2}{4}\left(\frac{9m}{n} + \frac{24}{n\varepsilon_1^2} + \frac{24}{n\varepsilon_1^2} + \frac{64}{mn\varepsilon_1^2\varepsilon_2^2}\right) \qquad (19)$$

$$= \frac{\pi^2}{4}\left(\frac{24}{n\varepsilon_1^2} + \frac{24}{n\varepsilon_1^2} + \frac{80}{n\varepsilon_1\varepsilon_2}\right) \leq \frac{10\pi^2}{n}\left(\frac{1}{\varepsilon_1} + \frac{1}{\varepsilon_2}\right)^2.$$

In the penultimate equality, we use the choice $m = \frac{8}{\varepsilon_1 \varepsilon_2}$, which minimizes the expression in the previous line. The privacy constraints are satisfied by the Laplace mechanism and checking the sensitivity of the batch means.

2. It is straightforward to check that

$$\mathbb{E}[\widehat{\rho}_{\mathrm{NI}}^{(SG)}] = \mathbb{E}[XY\mathbb{1}(|X| \leq \lambda_1, |Y| \leq \lambda_2)]$$
$$= \rho - \mathbb{E}[XY\mathbb{1}(|X| > \lambda_1 \text{ or } |Y| > \lambda_2)].$$

We can thus bound the bias of the estimator $\widehat{\rho}_{\mathrm{NI}}^{(SG)}$ as:

$$\left|\mathbb{E}[\widehat{\rho}_{\mathrm{NI}}^{(SG)}] - \rho\right| \leq \mathbb{E}[|XY|\mathbb{1}(|X| > \lambda_1 \text{ or } |Y| > \lambda_2)]$$

$$\leq \left(\mathbb{E}|X|^3\right)^{\frac{1}{3}} \left(\mathbb{E}|Y|^3\right)^{\frac{1}{3}} \left(\mathbb{P}(|X| > \lambda_1) + \mathbb{P}(|Y| > \lambda_2)\right)^{\frac{1}{3}}$$

$$\lesssim \exp\left(-\frac{1}{3}\left\{\frac{\lambda_1^2}{\eta_1^2} \wedge \frac{\lambda_2^2}{\eta_2^2}\right\}\right) \tag{20}$$

where we use the fact that $X$ is $\eta_1$-subgaussian and $Y$ is $\eta_2$-subgaussian. At the same time,

$$\mathrm{Var}(\widehat{\rho}_{\mathrm{NI}}^{(SG)}) \leq \frac{m^2}{k}\left[\mathbb{E}(\bar{X}^{(1)} + Z_1^{(1)})^4\right]^{1/2}\left[\mathbb{E}(\bar{Y}^{(1)} + Z_2^{(1)})^4\right]^{1/2}$$

$$\lesssim \frac{m^2}{k}\left(\frac{1}{m} + \frac{\lambda_1^2}{m^2\varepsilon_1^2}\right)\left(\frac{1}{m} + \frac{\lambda_2^2}{m^2\varepsilon_2^2}\right) \lesssim \frac{1}{n}\left(\frac{\lambda_1}{\varepsilon_1} + \frac{\lambda_2}{\varepsilon_2}\right)^2$$

where in the last step we use the choice of $m = \frac{\lambda_1 \lambda_2}{\varepsilon_1 \varepsilon_2}$, which minimizes the expression in the previous step. Thus the MSE of $\widehat{\rho}_\lambda^{(P)}$ in estimating $\rho$ is given by:

$$\mathbb{E}|\widehat{\rho}_{\mathrm{NI}}^{(SG)} - \rho|^2 = \left(\mathbb{E}[\widehat{\rho}_{\mathrm{NI}}^{(SG)}] - \rho\right)^2 + \mathrm{Var}(\widehat{\rho}_{\mathrm{NI}}^{(SG)})$$

$$\lesssim \exp\left(-\frac{2}{3}\left\{\frac{\lambda_1^2}{\eta_1^2} \wedge \frac{\lambda_2^2}{\eta_2^2}\right\}\right) + \frac{1}{n}\left(\frac{\lambda_1}{\varepsilon_1} + \frac{\lambda_2}{\varepsilon_2}\right)^2.$$

We now choose $\lambda_1 = 2\eta_1\sqrt{\log(n)}$ and $\lambda_2 = 2\eta_2\sqrt{\log(n)}$ for some $\kappa > 0$. The bias bound from (20) then becomes:

$$\left|\mathbb{E}[\widehat{\rho}_{\mathrm{NI}}^{(SG)}] - \rho\right| \leq \frac{1}{n} \tag{21}$$

leading to the MSE bound

$$\mathbb{E}|\widehat{\rho}_{\mathrm{NI}}^{(SG)} - \rho|^2 \lesssim \exp(-2\log(n)) + \frac{\log(n)}{n\varepsilon_1^2} + \frac{\log(n)}{n\varepsilon_2^2} \lesssim \frac{\log(n)}{n}\left(\frac{\eta_1}{\varepsilon_1^2} + \frac{\eta_2}{\varepsilon_2^2}\right).$$

Once again the privacy constraints are satisfied by the Laplace mechanism and checking the sensitivity of the batch means.

3. We split the proofs for confidence interval coverage into the Gaussian and sub-Gaussian cases.

   (a) (Gaussian case) Note that $\widehat{\eta}_{XY}$ in (3) is an average of $k$ iid observations $T_j$ defined as follows:

   $$\widehat{\eta}_{XY} = \frac{1}{k}\sum_{j=1}^{k} T_j \quad \text{where } T_j := m(\bar{X}^{(j)} + Z_1^{(j)})(\bar{Y}^{(j)} + Z_2^{(j)})$$

   and

   $$\sigma_\eta^2 := \mathrm{Var}(T_j)$$
   $$= m^2\mathbb{E}[(\bar{X}^{(j)})^2(\bar{Y}^{(j)} + Z_2^{(j)})^2] + \frac{8}{\varepsilon_1^2}\mathrm{Var}(\bar{Y}^{(j)} + Z_2^{(j)}) - (\mathbb{E}[\widehat{\eta}_{XY}])^2$$
   $$= m^2\mathbb{E}[(\bar{X}^{(j)})^2(\bar{Y}^{(j)})^2] - (\mathbb{E}[\widehat{\eta}_{XY}])^2 + m\mathrm{Var}(Z_2^{(j)}) + \frac{8}{\varepsilon_1^2}\left(\frac{1}{m} + \frac{8}{m^2\varepsilon_2^2}\right)$$
   $$= \left(\frac{m-1}{m}\right)^2[1 + (\mathbb{E}[\widehat{\eta}_{XY}])^2] + \frac{1}{m} + \frac{8}{m}\left(\frac{1}{\varepsilon_1^2} + \frac{1}{\varepsilon_2^2}\right) + \frac{64}{m^2\varepsilon_1^2\varepsilon_2^2}.$$

where the last equality follows by expanding the squares of iid averages in the first term. Thus we have

$$\frac{\sqrt{k}(\widehat{\eta}_{XY} - \mathbb{E}(\widehat{\eta}_{XY}))}{\sigma_\eta} \xrightarrow{d} N(0, 1) \quad \text{as } k \to \infty,$$

and thus by delta method, $\widehat{\rho}_{\mathrm{NI}}^{(G)} = \sin(\pi \widehat{\eta}_{XY}^{(P)}/2)$ satisfies:

$$\frac{\sqrt{k}(\widehat{\rho}_{\mathrm{NI}}^{(G)} - \rho)}{(\pi/2)\sigma_\eta \sqrt{1 - \rho^2}} \xrightarrow{d} N(0, 1) \quad \text{as } k \to \infty.$$

Here we used the fact that $\sin(\pi \mathbb{E}[\widehat{\eta}_{XY}]/2) = \rho$. To estimate $\sigma_\eta^2$ we use the sample variance of $T_j$:

$$S_\eta^2 := \frac{1}{k} \sum_{j=1}^{k} (T_j - \bar{T})^2.$$

Note that $S_\eta^2$ is constructed from $(\varepsilon_1, \varepsilon_2)$-DP statistics $T_j$, and thus $S_\eta^2$ is also differentially private. By standard calculations,

$$\mathbb{E}(S_\eta^2 - \sigma_\eta^2)^2 = O\left(\frac{1}{k}\right)$$

where we use our choice $= 8/(\varepsilon_1 \varepsilon_2)$ and $k = n/m$, and thus by Slutsky's theorem, we then have

$$\frac{\sqrt{k}(\widehat{\rho}_{\mathrm{NI}}^{(G)} - \rho)}{(\pi/2)S_\eta \sqrt{1 - (\widehat{\rho}_{\mathrm{NI}}^{(G)})^2}} \xrightarrow{d} N(0, 1) \quad \text{as } k \to \infty.$$

We thus have asymptotically $(1 - \alpha)$ coverage confidence intervals:

$$\left(\widehat{\rho}_{\mathrm{NI}}^{(G)} - \frac{\pi S_\eta \sqrt{1 - (\widehat{\rho}^{(P)})^2}}{2\sqrt{k}} z_{1-\alpha/2}, \widehat{\rho}_{\mathrm{NI}}^{(G)} + \frac{\pi S_\eta \sqrt{1 - (\widehat{\rho}^{(P)})^2}}{2\sqrt{k}} z_{1-\alpha/2}\right).$$

(b) (sub-Gaussian case) Identical to what we observed for the case of Gaussian data, note that $\widehat{\rho}_{\mathrm{NI}}^{(SG)}$ in (7) is an average of $k$ iid observations $T_j$ defined as follows:

$$\widehat{\rho}_{\mathrm{NI}}^{(SG)} = \frac{1}{k} \sum_{j=1}^{k} T_j \quad \text{where } T_j := m(\bar{X}^{(j)} + Z_1^{(j)})(\bar{Y}^{(j)} + Z_2^{(j)})$$

and

$$\sigma_\rho^2 := \mathrm{Var}(T_j)$$
$$= m^2 \mathbb{E}[(\bar{X}^{(j)})^2 (\bar{Y}^{(j)} + Z_2^{(j)})^2] + \frac{8}{\varepsilon_1^2} \mathrm{Var}(\bar{Y}^{(j)} + Z_2^{(j)}) - (\mathbb{E}[\widehat{\rho}_{\mathrm{NI}}^{(SG)}])^2$$
$$= m^2 \mathbb{E}[(\bar{X}^{(j)})^2 (\bar{Y}^{(j)})^2] - (\mathbb{E}[\widehat{\rho}_{\mathrm{NI}}^{(SG)}])^2 + m \mathrm{Var}(Z_2^{(j)}) + \frac{8}{\varepsilon_1^2}\left(\frac{\mathrm{Var}([Y]_{\lambda_2})}{m} + \frac{8}{m^2 \varepsilon_2^2}\right)$$
$$= \left(\frac{m-1}{m}\right)^2 (\mathrm{Var}([X]_{\lambda_1}) \mathrm{Var}([Y]_{\lambda_2}) + (\mathbb{E}[\widehat{\rho}_{\mathrm{NI}}^{(SG)}])^2) + \frac{\mathbb{E}([X]_{\lambda_1}^4 [Y]_{\lambda_2}^4)}{m}$$
$$+ \frac{8}{m}\left(\frac{\mathrm{Var}([Y]_{\lambda_2})}{\varepsilon_1^2} + \frac{\mathrm{Var}([X]_{\lambda_1})}{\varepsilon_2^2}\right) + \frac{64}{m^2 \varepsilon_1^2 \varepsilon_2^2}.$$

where the last equality follows by expanding the squares of iid averages in the first term of the previous line. Thus we have

$$\frac{\sqrt{k}(\widehat{\rho}_{\mathrm{NI}}^{(SG)} - \mathbb{E}(\widehat{\rho}_{\mathrm{NI}}^{(SG)}))}{\sigma_\rho} \xrightarrow{d} N(0, 1) \quad \text{as } k \to \infty,$$

To estimate $\sigma_\rho^2$ we use the sample variance of $T_j$:

$$S_\rho^2 := \frac{1}{k}\sum_{j=1}^k (T_j - \bar{T})^2.$$

Note that $S_\rho^2$ is constructed from $(\varepsilon_1, \varepsilon_2)$-DP statistics $T_j$, and thus $S_\rho^2$ is also differentially private. By standard calculations,

$$\mathbb{E}(S_\rho^2 - \sigma_\rho^2)^2 = O\left(\frac{1}{k}\right)$$

where we use our choice $m = 4\eta_1\eta_2(\log(n))/(\varepsilon_1\varepsilon_2)$ and $k = n/m$, and thus by Slutsky's theorem, along with the asymptotically vanishing bias from (21) we then have

$$\frac{\sqrt{k}(\widehat{\rho}_{\mathrm{NI}}^{(SG)} - \rho)}{S_\rho} \xrightarrow{d} N(0,1) \quad \text{as } k \to \infty.$$

We thus have an asymptotically $(1-\alpha)$ coverage confidence interval:

$$\left(\widehat{\rho}_{\mathrm{NI}}^{(SG)} - \frac{S_\rho}{\sqrt{k}}z_{1-\alpha/2}, \widehat{\rho}_{\mathrm{NI}}^{(SG)} + \frac{S_\rho}{\sqrt{k}}z_{1-\alpha/2}\right).$$

$\square$

*Proof of Theorem 3.1.* We separate the proofs of the two statements as follows.

1. To derive the MSE of the interactive correlation estimator for Gaussian data, we first calculate from (9):

$$\mathbb{E}\left[\widehat{\eta}_{XY,\mathrm{int}} + Z - (2\mathbb{P}(XY > 0) - 1)\right]^2$$
$$= \mathrm{Var}(\widehat{\eta}_{XY,\mathrm{int}} + Z) + (\mathbb{E}[\widehat{\eta}_{XY,\mathrm{int}}] - (2\mathbb{P}(XY > 0) - 1))^2$$
$$= \frac{4e^{\varepsilon_1}}{n(e^{\varepsilon_1} - 1)^2} + \frac{4(e^{\varepsilon_1} + 1)^2}{n^2(e^{\varepsilon_1} - 1)^2\varepsilon_2^2} + 0$$
$$\leq \frac{4}{n(\varepsilon_1 \wedge 1)^2} + \frac{25}{n^2(\varepsilon_1 \wedge 1)^2\varepsilon_2^2}.$$

Consequently,

$$\mathbb{E}(\widehat{\rho}_{\mathrm{INT}}^{(G)} - \rho)^2$$
$$= \frac{\pi^2(1-\rho^2)}{4}\mathbb{E}\left[\widehat{\eta}_{XY,\mathrm{int}}^{(\mathrm{P})} - (2\mathbb{P}(XY > 0) - 1)\right]^2 + \frac{\pi^4}{2}\mathbb{E}\left[\widehat{\eta}_{XY,\mathrm{int}}^{(\mathrm{P})} - (2\mathbb{P}(XY > 0) - 1)\right]^4$$
$$\leq \frac{\pi^2(1-\rho^2) + 1}{n(\varepsilon_1 \wedge 1)^2} + \frac{25\pi^2(1-\rho^2) + 1}{4n^2(\varepsilon_1 \wedge 1)^2\varepsilon_2^2}$$

whenever $n(\varepsilon_1 \wedge 1)$ is sufficiently large.

2. To derive the MSE of the interactive correlation estimator for sub-Gaussian data, we take the following approach. It is straightforward to check that

$$\mathbb{E}[\widehat{\rho}_{\mathrm{INT}}^{(SG)}] = \mathbb{E}\left([([X]_{\lambda_1} + Z)Y]_{\lambda_2}\right)$$
$$= \mathbb{E}\left((([X]_{\lambda_1} + Z)Y) - \mathbb{E}[([X]_{\lambda_1} + Z)Y\mathbb{1}(|([X]_{\lambda_1} + Z)Y| > \lambda_2)]\right.$$
$$= \rho - \mathbb{E}[XY\mathbb{1}(|X| > \lambda_1)] - \mathbb{E}[([X]_{\lambda_1} + Z)Y\mathbb{1}(|([X]_{\lambda_1} + Z)Y| > \lambda_2)].$$

We next have

$$\mathbb{P}(|([X]_{\lambda_1} + Z)Y| \geq \lambda_2) \leq \mathbb{P}(|Y| \gtrsim \sqrt{\log(n)}) + \mathbb{P}(|([X]_{\lambda_1} + Z)| > \lambda_2/\sqrt{\log(n)})$$
$$\lesssim \frac{1}{n} + \exp\left(-\frac{1}{2}\left\{\frac{\lambda_2^2}{(\log(n))\eta_1^2} \wedge \frac{\varepsilon_1\lambda_2}{2\lambda_1\sqrt{\log(n)}}\right\}\right) \quad (22)$$

the fact that $X$ is $\eta_1$-subgaussian and $Y$ is $\eta_2$-subgaussian. We can then bound the bias of the estimator $\widehat{\rho}_{\mathrm{INT}}^{(SG)}$ as:

$$\left|\mathbb{E}[\widehat{\rho}_{\mathrm{INT}}^{(SG)}] - \rho\right|$$

$$\leq \mathbb{E}[|XY|\mathbb{1}(|X| > \lambda_1)] + \mathbb{E}[(|XY| + |ZY|)(\mathbb{1}(|([X]_{\lambda_1} + Z)Y| > \lambda_2))]$$

$$\leq (\mathbb{E}[|XY|^2])^{1/2}[\mathbb{P}(|X| > \lambda_1)]^{1/2}$$

$$\quad + (\mathbb{E}|XY|^2 + \mathbb{E}|ZY|^2)^{1/2}[\mathbb{P}(|Y| \geq 2\sqrt{\log(n)}) + \mathbb{P}(|([X]_{\lambda_1} + Z)| > \lambda_2/\sqrt{\log(n)})]^{1/2}$$

$$\lesssim \frac{1}{n} \tag{23}$$

where we use (22) with

$$\lambda_1 = 2\eta_1\sqrt{\log(n)} \quad \text{and} \quad \lambda_2 = 4(\eta_2 \vee 1)(\log(n))^2/(\varepsilon_1 \wedge 1)$$

and hence the variance becomes

$$\mathrm{Var}(\widehat{\rho}_{\mathrm{INT}}^{(SG)}) \leq \frac{\mathrm{Var}(([X]_{\lambda_1} + Z_1)Y)}{n} + \frac{4\lambda_2^2}{n^2\varepsilon_2^2}$$

$$\leq \frac{\mathrm{Var}(XY)}{n} + \frac{4\lambda_1^2}{n\varepsilon_1^2} + \frac{4\lambda_2^2}{n^2\varepsilon_2^2}$$

$$= \frac{\mathrm{Var}(XY)}{n} + \frac{16\eta_1^2(\log(n))}{n\varepsilon_1^2} + \frac{64(\eta_2^2 \vee 1)(\log(n))^4}{n^2\varepsilon_2^2(\varepsilon_1 \wedge 1)^2}.$$

3. We split the proofs for confidence interval coverage into the Gaussian and sub-Gaussian cases.

   (a) (Gaussian case) From (9) we write:

   $$\widehat{\eta}_{XY,\mathrm{int}} + Z = \frac{\exp(\varepsilon_1) + 1}{\exp(\varepsilon_1) - 1} \times \frac{1}{n}\sum_{i=1}^{n} T_i + Z =: \frac{\exp(\varepsilon_1) + 1}{\exp(\varepsilon_1) - 1}(\bar{T} + Z_2)$$

   where $T_i = (2S_i - 1)\,\mathrm{sign}(X_i)\,\mathrm{sign}(Y_i)$ and $Z_2 \sim \mathrm{Laplace}\left(0, \frac{2}{n\varepsilon_2}\right)$. Let us define:

   $$\sigma_\eta^2 := \mathrm{Var}(T_i) = 1 - \left(\frac{\exp(\varepsilon_1) - 1}{\exp(\varepsilon_1) + 1}\right)^2 (2\mathbb{P}(XY > 0) - 1)^2$$

   for which we have the consistent estimator:

   $$\widehat{\sigma}_\eta^2 := 1 - \left(\frac{\exp(\varepsilon_1) - 1}{\exp(\varepsilon_1) + 1}\right)^2 (\widehat{\eta}_{XY,\mathrm{int}})^2.$$

   We recall that $\mathbb{E}[\widehat{\eta}_{XY,\mathrm{int}}] = 2\mathbb{P}(XY > 0) - 1$ and thus by the Berry Esseen limit theorem on $T_i$,

   $$\sup_x \left|\mathbb{P}\left(\frac{\sqrt{n}}{\sigma_\eta}\left(\frac{\exp(\varepsilon_1) - 1}{\exp(\varepsilon_1) + 1}\right)\left(\widehat{\eta}_{XY,\mathrm{int}}^{(\mathrm{P})} - \mathbb{P}(|XY| > 0)\right) \leq x\right)\right.$$
   $$\left. - \mathbb{P}(Z_{XY} + Z_2' \leq x)\right| \leq \frac{C}{\sigma_\eta^3\sqrt{n}} \tag{24}$$

   for a numerical constant $C > 0$. Here

   $$Z_{XY} \sim N(0,1) \quad \text{and} \quad Z_2' \sim \mathrm{Laplace}\left(0, \frac{2}{\sqrt{n}\sigma_\eta\varepsilon_2}\right).$$

   and thus by the delta method,

   $$\sup_x \left|\mathbb{P}\left(\frac{2\sqrt{n}}{\pi\sqrt{1 - \rho^2}\sigma_\eta}\left(\frac{\exp(\varepsilon_1) - 1}{\exp(\varepsilon_1) + 1}\right)\left(\widehat{\rho}_{\mathrm{INT}}^{(G)} - \rho\right) \leq x\right)\right.$$
   $$\left. - \mathbb{P}(Z_{XY} + Z_2' \leq x)\right| \leq \frac{C}{\sigma_\eta^3\sqrt{n}}. \tag{25}$$

To derive the confidence intervals we make two separate cases:

**Case 1:** $((\sqrt{n}\varepsilon_2)^{-1} \to c)$ In the first case we consider $(\sqrt{n}\varepsilon_2)^{-1} \to c$ for a finite constant $c \geq 0$. In this case we have the confidence interval

$$\left( \widehat{\rho}_{\text{INT}}^{(G)} \mp \frac{\pi \widehat{\sigma}_\eta \sqrt{1 - (\widehat{\rho}_{\text{INT}}^{(G)})^2}}{2\sqrt{n}} \left( \frac{\exp(\varepsilon_1) + 1}{\exp(\varepsilon_1) - 1} \right) F^{-1}(1 - \alpha/2) \right)$$

where for any $x \in \mathbb{R}$ we define

$$F(x) := \mathbb{P}(Z_{XY} + c_* Z_{\text{Lap}} \leq x) \text{ where } c_* = \lim_{n \to \infty} \frac{2}{\sqrt{n}\sigma_\eta \varepsilon_2} \text{ and } Z_{\text{Lap}} \sim \text{Laplace}(0, 1).$$

The above is a valid confidence interval when $\lim_{n \to \infty} \frac{2}{\sqrt{n}\sigma_\eta \varepsilon_2} = c_*$ for some finite $c_* \geq 0$. This is no longer the case when $\sqrt{n}\varepsilon_2 \to 0$ as $n \to \infty$.

**Case 2:** $(\sqrt{n}\varepsilon_2 \to 0)$ In this case, (24) and (25) imply that we have the asymptotic convergence:

$$\frac{n\varepsilon_2}{\pi\sqrt{1 - \rho^2}} \left( \frac{\exp(\varepsilon_1) - 1}{\exp(\varepsilon_1) + 1} \right) \left( \widehat{\rho}_{\text{INT}}^{(G)} - \rho \right) \xrightarrow{d} \text{Laplace}(0, 1)$$

leading to the asymptotically $(1 - \alpha)$ coverage confidence interval

$$\left( \widehat{\rho}_{\text{INT}}^{(G)} \pm \frac{\pi \sqrt{1 - (\widehat{\rho}_{\text{int}}^{(P)})^2}}{n\varepsilon_2} \left( \frac{\exp(\varepsilon_1) + 1}{\exp(\varepsilon_1) - 1} \right) \log(\alpha) \right)$$

where the width of the CI is determined by the $\alpha$-th quantiles of the $\text{Laplace}(0, 1)$ distribution.

(b) (sub-Gaussian case) Note that

$$\widehat{\rho}_{\text{INT}}^{(SG)} = \frac{1}{n} \sum_{i=1}^{n} T_i + Z_2$$

where $T_i = [([X_i]_{\lambda_1} + Z_{1i})Y_i]_{\lambda_2}$ are iid random variables. Thus by the Berry Esseen theorem,

$$\sup_{x \in \mathbb{R}} \left| \mathbb{P}\left( \frac{\sqrt{n}(\widehat{\rho}_{\text{INT}}^{(SG)} - \mathbb{E}[\widehat{\rho}_{\text{INT}}^{(SG)}])}{\sigma_\rho} \leq x \right) - \mathbb{P}\left( Z_{XY} + Z_2' \leq x \right) \right| \leq \frac{C(\log(n))^{7.5}}{\sigma_\rho^3 \sqrt{n}}$$

$$(26)$$

where $Z_{XY} \sim N(0, 1)$, $Z_2' \sim \text{Laplace}\left( 0, \frac{2\lambda_2}{\sqrt{n}\sigma_\rho \varepsilon_2} \right)$ and

$$\sigma_\rho^2 := \text{Var}(([X]_{\lambda_1} + Z_1)Y).$$

As before, we now make two cases to derive the confidence intervals.

**Case 1:** $((\sqrt{n}\varepsilon_2/\lambda_2)^{-1} \to c)$ In the first case we consider $(\sqrt{n}\varepsilon_2/\lambda_2)^{-1} \to c$ for a finite constant $c \geq 0$. In this case (23) and (26) imply that we have the confidence interval

$$\left( \widehat{\rho}_{\text{INT}}^{(SG)} - \frac{\widehat{\sigma}_\rho}{\sqrt{n}} F^{-1}(1 - \alpha/2), \widehat{\rho}_{\text{INT}}^{(SG)} + \frac{\widehat{\sigma}_\rho}{\sqrt{n}} F^{-1}(1 - \alpha/2) \right)$$

where

$$\widehat{\sigma}_\rho^2 = \frac{1}{n} \sum_{i=1}^{n} (T_i - \bar{T})^2,$$

the sample variance of $T_i$, is an $\varepsilon_1$-DP consistent estimator for $\sigma_\rho^2$. Moreover, as before for any $x \in \mathbb{R}$ we define

$$F(x) := \mathbb{P}(Z_{XY} + c_* Z_{\text{Lap}} \leq x) \text{ where } c_* = \lim_{n \to \infty} \frac{2\lambda_2}{\sqrt{n}\sigma_\rho \varepsilon_2} \text{ and } Z_{\text{Lap}} \sim \text{Laplace}(0, 1).$$

The above is a valid confidence interval when $\lim_{n \to \infty} \frac{2\lambda_2}{\sqrt{n}\sigma_\rho \varepsilon_2} = c_*$ for some finite $c_* \geq 0$. This is no longer the case when $\sqrt{n}\varepsilon_2/\lambda_2 \to 0$ as $n \to \infty$.

**Case 2:** $(\sqrt{n}\varepsilon_2/\lambda_2 \to 0)$ In this case, (23) and (26) imply that we have the asymptotic convergence:

$$\frac{n\varepsilon_2}{2\lambda_2}\left(\widehat{\rho}_{\mathrm{INT}}^{(SG)} - \rho\right) \xrightarrow{d} \mathrm{Laplace}(0,1)$$

leading to the asymptotically $(1-\alpha)$ coverage confidence interval

$$\left(\widehat{\rho}_{\mathrm{INT}}^{(SG)} + \frac{\lambda_2}{n\varepsilon_2}\log(\alpha), \widehat{\rho}_{\mathrm{INT}}^{(SG)} - \frac{\lambda_2}{n\varepsilon_2}\log(\alpha)\right)$$

where the width of the CI is determined by the $\alpha$-th quantiles of the $\mathrm{Laplace}(0,1)$ distribution.

$\square$

## B.2   Proofs of lower bound results

*Proof of Theorem 4.1 .* Fix any non–interactive $(\varepsilon_1, \varepsilon_2, \delta_1, \delta_2)$–DP protocol with transcript $T = (T_1, T_2)$, and let $P_\rho$ denote the law of $T$ when $(X_{i1}, X_{i2})_{i=1}^n \overset{\mathrm{i.i.d.}}{\sim} \mathcal{N}\left(\mathbf{0}, \left(\begin{smallmatrix} 1 & \rho \\ \rho & 1 \end{smallmatrix}\right)\right)$. We check that $f(x) = x\log(1/x)$ is an increasing function of $x$ whenever $x \in (0, \exp(-1))$. Thus $\delta_k = o(n^{-1-\omega})$ implies $\delta_k \log(1/\delta_k) = o(n^{-1}) = o(\varepsilon_k^2)$. The second inequality follows from the fact that $n\varepsilon_k^2 \to \infty$. Invoking Lemma 2 gives, at $\rho = 0$,

$$I_F(T; 0) \leq \frac{8}{\pi}\left(n\varepsilon_1^2 \wedge n\varepsilon_2^2\right). \tag{27}$$

**Step 1. Prior supported in a small neighborhood of** $0$**.** Let $\mathcal{J} = [-L/2, L/2]$ with $L \leq 2c_0$ and center $\rho_0 = 0$. Define the cosine–squared prior on $\mathcal{J}$:

$$\lambda(\rho) = \frac{2}{L}\lambda_0\left(\frac{2(\rho - \rho_0)}{L}\right), \qquad \lambda_0(x) = \begin{cases} \cos^2(\pi x/2), & |x| \leq 1, \\ 0, & \text{otherwise.} \end{cases}$$

This prior satisfies the well–known identity (see, e.g., Tsybakov [2009])

$$I_F(\lambda) = \int_{\mathcal{J}} \frac{\lambda'(\rho)^2}{\lambda(\rho)}\,d\rho = \left(\frac{2\pi}{L}\right)^2. \tag{28}$$

**Step 2. Prior–averaged information of the transcript.** By (16) with $\rho_0 = 0$ and $|\rho| \leq L/2 \leq c_0$,

$$I_F(T; \rho) \leq (1 + C_\eta)\,I_F(T; 0).$$

Therefore

$$\mathbb{E}_\rho\big[I_F(T; \rho)\big] = \int_{\mathcal{J}} I_F(T; \rho)\,\lambda(\rho)\,d\rho \leq (1 + C_\eta)\,I_F(T; 0). \tag{29}$$

**Step 3. Van Trees inequality.** Applying Lemma 1 with parameter $\rho$, likelihood $P_\rho$, and prior $\lambda$, we obtain the Bayes risk lower bound

$$\mathbb{E}\big[(\widehat{\rho}(T) - \rho)^2\big] \geq \frac{1}{\mathbb{E}_\rho[I_F(T; \rho)] + I_F(\lambda)} \overset{(29),(28)}{\geq} \frac{1}{(1 + C_\eta)\,I_F(T; 0) + \left(2\pi/L\right)^2}.$$

Using (27) and writing $A := n\varepsilon_1^2 \wedge n\varepsilon_2^2$,

$$\mathcal{R}_{\mathrm{Bayes}}(\lambda) := \inf_{\widehat{\rho}} \mathbb{E}[(\widehat{\rho} - \rho)^2] \geq \frac{1}{c_1 A + (2\pi/L)^2}, \qquad c_1 := \frac{8}{\pi}(1 + C_\eta). \tag{30}$$

**Step 4. Choice of $L$ and consequence.** To minimize the denominator in (30) we take the largest admissible support, $L = 2c_0$, yielding

$$\mathcal{R}_{\text{Bayes}}(\lambda) \;\geq\; \frac{1}{c_1 A \;+\; (\pi/c_0)^2}.$$

Since the minimax risk dominates the Bayes risk for any prior,

$$\inf_{\widehat{\rho}} \sup_{\rho \in [-1,1]} \mathbb{E}_\rho\big[(\widehat{\rho} - \rho)^2\big] \;\geq\; \mathcal{R}_{\text{Bayes}}(\lambda) \;\geq\; \frac{1}{c_1 A \;+\; (\pi/c_0)^2}.$$

In particular, whenever $A \to \infty$ (our standing regime), the constant prior term is negligible and we obtain

$$\inf_{\widehat{\rho}} \sup_{\rho \in [-1,1]} \mathbb{E}_\rho\big[(\widehat{\rho} - \rho)^2\big] \;\gtrsim\; \frac{1}{n\varepsilon_1^2 \wedge n\varepsilon_2^2}.$$

**Step 5. Classical $(1/n)$ term.** Additionally the non–private parametric difficulty contributes an additional $\Omega(1/n)$ term (e.g. by repeating the bound above without privacy constraints and with $A \asymp n$ near $\rho = 0$). Combining the terms yields

$$\inf_{\widehat{\rho} \in \text{NI}(\varepsilon_1, \varepsilon_2, \delta)} \sup_{\rho \in [-1,1]} \mathbb{E}_\rho\big[(\widehat{\rho} - \rho)^2\big] \;\gtrsim\; \frac{1}{n} \;+\; \frac{1}{n\varepsilon_1^2} \;+\; \frac{1}{n\varepsilon_2^2}. \hspace{1cm} \square$$

*Proof of Theorem 4.2.* Let $(i,j)$ be such that $\varepsilon_i \geq \varepsilon_j$. Since $\delta_i = o(n^{-1-\omega})$, we have $\delta_i \log(1/\delta_i) = o(n^{-1}) = o(\varepsilon_i^2)$, where the last equality follows from $n\varepsilon_i^2 \to \infty$. Similarly, $\delta_j = o(n^{-1-\omega})$ implies $\delta_j \log^2(1/\delta_j) = o(n^{-1}) = o(n\varepsilon_1^2\varepsilon_2^2)$, using that $n^2\varepsilon_1^2\varepsilon_2^2 \to \infty$. Hence, the conditions of Lemma 3 are met, so that $C_{\Pi,n}$ in (17) indeed represents the Fisher–information bound for the (interactive, one–way) protocol.

Throughout, we abbreviate

$$\varepsilon_{\max}^2 = \varepsilon_1^2 \vee \varepsilon_2^2, \qquad \varepsilon_{\min}^2 = \varepsilon_1^2 \wedge \varepsilon_2^2, \qquad C_{\Pi,n} = n\varepsilon_{\max}^2 \;\wedge\; n^2\varepsilon_1^2\varepsilon_2^2.$$

Let $\Pi \in \{\Pi_{1\to2}, \Pi_{2\to1}\}$ be any fixed one–way interactive DP protocol, and denote by $P_\rho$ the law of the full transcript under correlation $\rho$.

**Step 1. Local regularity of information and the prior.** By the standing regularity assumption (16), there are numerical constants $c_0 \in (0,1)$ and $C_\eta > 0$ such that

$$I_F(\Pi; \rho + \epsilon) = I_F(\Pi; \rho)\big(1 + \eta(\epsilon)\big), \qquad |\epsilon| < c_0, \quad \sup_{|\epsilon| < c_0} |\eta(\epsilon)| \leq C_\eta.$$

In particular, for $|\rho| \leq c_0$,

$$I_F(\Pi; \rho) \;\leq\; (1 + C_\eta)\, I_F(\Pi; 0). \tag{31}$$

We place on $\rho$ the cosine–squared prior supported on $\mathcal{J} = [-L/2, L/2]$ with $L \leq 2c_0$ and center 0 that the prior Fisher information is (see proof of Theorem 4.1 for details)

$$I_F(\lambda) = \int_{\mathcal{J}} \frac{\lambda'(\rho)^2}{\lambda(\rho)} \, d\rho = \Big(\frac{2\pi}{L}\Big)^2. \tag{32}$$

**Step 2. Prior–averaged information of the transcript.** By (31) and Lemma 3 (or its analogue for $\Pi_{2\to1}$),

$$\mathbb{E}_\rho\big[I_F(\Pi; \rho)\big] = \int_{\mathcal{J}} I_F(\Pi; \rho)\, \lambda(\rho)\, d\rho \;\leq\; (1 + C_\eta)\, I_F(\Pi; 0) \;\leq\; (1 + C_\eta)\, C_{\Pi,n}.$$

**Step 3. Van Trees inequality.** Applying Lemma 1 with parameter $\rho$, likelihood $P_\rho$, and prior $\lambda$, we obtain

$$\mathcal{R}_{\text{Bayes}}(\lambda) := \inf_{\widehat{\rho}} \mathbb{E}\big[(\widehat{\rho} - \rho)^2\big] \;\geq\; \frac{1}{\mathbb{E}_\rho[I_F(\Pi; \rho)] \;+\; I_F(\lambda)} \;\geq\; \frac{1}{c_1 C_{\Pi,n} \;+\; (2\pi/L)^2},$$

where $c_1 := 1 + C_\eta$ is an absolute constant. Choosing the largest admissible support $L = 2c_0$ gives

$$\mathcal{R}_{\text{Bayes}}(\lambda) \;\geq\; \frac{1}{c_1 C_{\Pi,n} \;+\; (\pi/c_0)^2}. \tag{33}$$

Since the minimax risk dominates the Bayes risk for every prior,

$$\inf_{\widehat{\rho}} \sup_{\rho \in [-1,1]} \mathbb{E}_\rho\big[(\widehat{\rho} - \rho)^2\big] \;\geq\; \mathcal{R}_{\text{Bayes}}(\lambda).$$

**Step 4. Extracting the two interactive terms.** By definition, $C_{\Pi,n} = \min\{A, B\}$ with

$$A := n\varepsilon_{\max}^2, \qquad B := n^2\varepsilon_1^2\varepsilon_2^2.$$

Hence $1/C_{\Pi,n} = \max\{1/A, 1/B\} \geq \frac{1}{2}(1/A + 1/B)$. Using (33) and the fact that the additive constant $(\pi/c_0)^2$ is negligible whenever $A \vee B \to \infty$, we obtain the privacy–induced contribution

$$\inf_{\widehat{\rho}} \sup_{\rho} \mathbb{E}_{\rho}\big[(\widehat{\rho} - \rho)^2\big] \gtrsim \frac{1}{n\varepsilon_{\max}^2} + \frac{1}{n^2\varepsilon_1^2\varepsilon_2^2}.$$

**Step 5. Baseline parametric term and conclusion.** Even without privacy constraints, estimating a correlation from $n$ i.i.d. Gaussian samples incurs risk $\Theta(1/n)$; hence

$$\inf_{\widehat{\rho} \in \mathrm{INT}(\varepsilon_1, \varepsilon_2, \delta)} \sup_{\rho \in [-1, 1]} \mathbb{E}_{\rho}\big[(\widehat{\rho} - \rho)^2\big] \gtrsim \frac{1}{n} + \frac{1}{n\varepsilon_{\max}^2} + \frac{1}{n^2\varepsilon_1^2\varepsilon_2^2},$$

which is the desired bound. $\qquad\qquad\qquad\qquad\qquad\qquad\qquad\qquad\qquad\qquad\qquad\quad\square$

## B.3   Proofs of Lemmas

In this section we provide proofs of lemmas used to prove the lower bound theorems.

*Proof of Lemma 2.* The main technical ingredient that goes into proving the minimax lower bound is obtaining a upper bound on the Fisher Information under the null i.e $\rho = 0$. Denote $\boldsymbol{Z} = (X_i, Y_i)_{i=1}^n$, it can be shown that the score function $S_\rho(\boldsymbol{Z})$ for the parameter $\rho$ under the null is given by

$$S_\rho(\boldsymbol{Z}) = \sum_{i=1}^n X_i Y_i \tag{34}$$

The fisher information under the null $I_F(T; 0)$ is given by

$$I_F(T; 0) = \mathbb{E}(\mathbb{E}(S_\rho(\boldsymbol{Z}) \mid T)^2) \tag{35}$$

The fisher info under null can be expressed as

$$I_F(T; 0) = \mathbb{E}\left[\left(\sum_{i=1}^n \mathbb{E}(X_i Y_i \mid T)\right)^2\right]$$

$$= \mathbb{E}\left[\left(\sum_{i=1}^n \mathbb{E}(X_i \mid T_1)\mathbb{E}(Y_i \mid T_2)\right)^2\right]$$

$$= \sum_{i=1}^n \sum_{j=1}^n \mathbb{E}\left[\mathbb{E}(X_i \mid T_1)\mathbb{E}(Y_i \mid T_2)\mathbb{E}(X_j \mid T_1)\mathbb{E}(Y_j \mid T_2)\right]$$

where we have used the fact that $X_i \perp Y_i \mid T$, $X_i \perp T_2 \mid T_1$ and $Y_i \perp T_1 \mid T_2$ in the second line. We now have that

$$I_F(T; 0) = \sum_{i=1}^n \sum_{j=1}^n \mathbb{E}\left[\mathbb{E}(X_i|T_1)\mathbb{E}(X_j|T_1)\right] \mathbb{E}\left[\mathbb{E}(Y_i|T_2)\mathbb{E}(Y_j|T_2)\right]$$

where in we use the fact that under the null $T_1 \perp T_2$. Define to matrices $M_X$ and $M_Y$ such that

$$(M_X)_{ij} = \mathbb{E}\left[\mathbb{E}(X_i \mid T_1)\mathbb{E}(X_j \mid T_1)\right] \text{ and } (M_Y)_{ij} = \mathbb{E}\left[\mathbb{E}(Y_i \mid T_2)\mathbb{E}(Y_j \mid T_2)\right]$$

then we have that $I_F(T; 0)] = \mathrm{tr}(M_X^\top M_Y)$. Using Lemma 6 we have that $I_F(T; 0)] \leq \mathrm{tr}(M_X)\|M_Y\|_2$ where $\|.\|_2$ is the spectral norm . Next let us bound $\mathrm{tr}(M_X)$. Note that we can rewrite $M_X$ as

$$M_X = \mathbb{E}\left(\mathbb{E}(\boldsymbol{X} \mid T_1)\mathbb{E}(\boldsymbol{X} \mid T_1)^\top\right) \tag{36}$$

where $X$ is the data vector $(X_i)_{i=1}^n$ and $\mathbb{E}(\boldsymbol{X} \mid T)$ is the vector $(\mathbb{E}(X_i \mid T))_{i=1}^n$. Hence

$$\operatorname{tr}(M_X) \le \operatorname{tr}\left(\mathbb{E}\left(\mathbb{E}(\boldsymbol{X} \mid T_1)\mathbb{E}(\boldsymbol{X} \mid T_1)^\top\right)\right)$$

$$= \mathbb{E}\|\mathbb{E}(\boldsymbol{X} \mid T_1)\|_2^2$$

$$= \sum_{i=1}^n \mathbb{E}(\mathbb{E}(X_i \mid T_1))^2$$

Using Lemma 5 we have that $\operatorname{tr}(M_X) \le n\frac{2}{\pi}\left(\frac{e^{\varepsilon_1}-e^{-\varepsilon_1}}{2}\right)^2$. For bounding $\|M_Y\|_2$ we can either bound by $\operatorname{tr}(M_Y)$ which implies by the previous argument that $\|M_Y\|_2 \le n\varepsilon_2^2$ or using contraction of the conditional expectation i.e.

$$M_X = \mathbb{E}(\mathbb{E}(\boldsymbol{X} \mid T)\mathbb{E}(\boldsymbol{X} \mid T)^\top) \preceq \mathbb{E}(\boldsymbol{X}\boldsymbol{X}^\top) = I_n$$

which implies $\|M_X\|_2 \le 1$. Putting everything together we have that

$$I_F(T;0) \le \operatorname{tr}(M_X)\|M_Y\|_2 \wedge \operatorname{tr}(M_Y)\|M_X\|_2$$

$$\le n\frac{2}{\pi}\left(\frac{e^{\varepsilon_1}-e^{-\varepsilon_1}}{2}\right)^2 \wedge n\frac{2}{\pi}\left(\frac{e^{\varepsilon_1}-e^{-\varepsilon_1}}{2}\right)^2.$$

Using the fact that $e^x - 1 \le 2x$ for $0 < x < 1$ we have that

$$I_F(T;0) \le \frac{8}{\pi}\left(n\varepsilon_1^2 \wedge n\varepsilon_2^2\right).$$

$\square$

*Proof of Lemma 3.* Denote $\boldsymbol{Z} = (X_i, Y_i)_{i=1}^n$, it can be shown that the score function $S_\rho(\boldsymbol{Z})$ for the parameter $\rho$ under the null is given by

$$S_\rho(\boldsymbol{Z}) = \sum_{i=1}^n X_i Y_i \tag{37}$$

The fisher information under the null $I_F(T;0)$ is given by

$$I_F(T;0) = \mathbb{E}(\mathbb{E}(S_\rho(\boldsymbol{Z}) \mid T)^2) \tag{38}$$

The fisher info under null can be expressed as

$$I_F(T;0) = \mathbb{E}\left[\left(\sum_{i=1}^n \mathbb{E}(X_i Y_i \mid T)\right)^2\right]$$

$$= \mathbb{E}\left[\left(\sum_{i=1}^n \mathbb{E}(X_i \mid T_1)\mathbb{E}(Y_i \mid T_1, T_2)\right)^2\right]$$

$$= \mathbb{E}\left[\left(\sum_{i=1}^n \mathbb{E}(\mathbb{E}(X_i \mid T_1)Y_i \mid T_1, T_2)\right)^2\right] \tag{39}$$

where we have used the fact that $X_i \perp Y_i \mid T$, $X_i \perp T_2 \mid T_1$ in the second line. Using the fact that $\mathbb{E}(\mathbb{E}(A|B)^2) \le \mathbb{E}A^2$ we have that

$$I_F(T;0) = \mathbb{E}\left[\left(\sum_{i=1}^n \mathbb{E}(\mathbb{E}(X_i \mid T_1)Y_i \mid T_1, T_2)\right)^2\right] \le \mathbb{E}\left[\left(\sum_{i=1}^n \mathbb{E}(\mathbb{E}(X_i \mid T_1)Y_i)\right)^2\right]$$

Hence expanding the sum of squares we have that

$$I_F(T;0) = \sum_{i=1}^{n}\sum_{j=1}^{n}\mathbb{E}\left[\mathbb{E}(X_i \mid T_1)Y_i\mathbb{E}(X_j \mid T_1)Y_j\right]$$

$$= \sum_{i=1}^{n}\mathbb{E}\left[\mathbb{E}(X_i \mid T_1)^2 Y_i^2\right]$$

$$= \sum_{i=1}^{n}\mathbb{E}\left[\mathbb{E}(X_i \mid T_1)^2\right] \le n\left(\frac{e^{\varepsilon_1}-e^{-\varepsilon_1}}{2}\right)^2$$

where we used the fact that $Y_i \perp Y_j, T_1$ and $\mathbb{E}Y_i = 0$, $\mathbb{E}Y_i^2 = 1$ in the second and third line. The last inequality above follows from Lemma 5.

Following (39) we can write

$$I_F(T;0) = \sum_{k=1}^{n}\mathbb{E}\left[\mathbb{E}(X_k \mid T_1)Y_k\left(\sum_{i=1}^{n}\mathbb{E}(\mathbb{E}(X_i \mid T_1)Y_i \mid T_1, T_2)\right)\right] \tag{40}$$

Let us call $G_k = \mathbb{E}(X_k \mid T_1)Y_k\left(\sum_{i=1}^{n}\mathbb{E}(\mathbb{E}(X_i \mid T_1)Y_i \mid T_1, T_2)\right)$ and $G'_k = \mathbb{E}(X_k \mid T_1)Y_k\left(\sum_{i=1}^{n}\mathbb{E}(\mathbb{E}(X_i \mid T_1)Y_i \mid T_1, T'_2)\right)$. Also note that $\mathbb{E}G'_k = 0$ since $\mathbb{E}Y_k = 0$ and $Y_k \perp T_1, T'_2$. Now following a similar argument as in (46) we get that

$$\mathbb{E}G_k \le \left(\frac{e^{\varepsilon_2}-e^{-\varepsilon_2}}{2}\right)\mathbb{E}|G'_k| + 2\delta_2 M + \int_M^{\infty}\mathbb{P}(|G_k|\ge t)dt + \int_M^{\infty}\mathbb{P}(|G'_k|\ge t)dt \tag{41}$$

Note that

$$\mathbb{E}|G'_k| = \mathbb{E}\left|\mathbb{E}(X_k \mid T_1)Y_k\left(\sum_{i=1}^{n}\mathbb{E}(\mathbb{E}(X_i \mid T_1)Y_i \mid T_1, T'_2)\right)\right|$$

$$= \sqrt{\frac{2}{\pi}}\mathbb{E}\left|\mathbb{E}(X_k \mid T_1)\left(\sum_{i=1}^{n}\mathbb{E}(\mathbb{E}(X_i \mid T_1)Y_i \mid T_1, T'_2)\right)\right|$$

$$\le \sqrt{\frac{2}{\pi}}\sqrt{\mathbb{E}(\mathbb{E}(X_k \mid T_1)^2)}\sqrt{\mathbb{E}\left[\left(\sum_{i=1}^{n}\mathbb{E}(\mathbb{E}(X_i \mid T_1)Y_i \mid T_1, T'_2)\right)^2\right]}$$

$$\le \sqrt{\frac{2}{\pi}}\left(\frac{e^{\varepsilon_1}-e^{-\varepsilon_1}}{2}\wedge 1\right)\sqrt{I_F(T;0)}.$$

The last line follows since $(T_1, T'_2)\stackrel{d}{=}(T_1, T_2)$ and the fact that $\mathbb{E}(\mathbb{E}(X_k \mid T_1)^2)\le \mathbb{E}X_k^2 = 1$. Using the fact that $I_F(T;0) = \sum_k \mathbb{E}G_k$ and putting everything together we have that

$$I_F(T;0) \le n\left(\frac{e^{\varepsilon_2}-e^{-\varepsilon_2}}{2}\right)\sqrt{\frac{2}{\pi}}\left(\frac{e^{\varepsilon_1}-e^{-\varepsilon_1}}{2}\wedge 1\right)\sqrt{I_F(T;0)} \tag{42}$$

$$+ 2n\delta_2 M + n\int_M^{\infty}\mathbb{P}(|G_k|\ge t)dt + n\int_M^{\infty}\mathbb{P}(|G'_k|\ge t)dt \tag{43}$$

Set

$$M = 64\left(\log\frac{8}{\delta_2}\right)^2$$

in Lemma 4, to obtain

$$\int_M^{\infty}\mathbb{P}(|G_k|\ge t)\,dt \le 16\left(8\log(8/\delta_2)+4\right)(\delta_2/8)^2 \le \delta_2.$$

We can similarly show that

$$\int_M^{\infty}\mathbb{P}(|G'_k|\ge t)dt \le \delta_2.$$

Putting everything together we have that

$$I_F(T;0) \le n\left(\frac{e^{\varepsilon_2} - e^{-\varepsilon_2}}{2}\right)\sqrt{\frac{2}{\pi}}\left(\frac{e^{\varepsilon_1} - e^{-\varepsilon_1}}{2}\right)\sqrt{I_F(T;0)} \tag{44}$$

$$+ 2n\delta_2 64\left(\log\frac{8}{\delta_2}\right)^2 + 2n\delta_2 \tag{45}$$

If $\sqrt{I_F(T;0)} \le n\sqrt{\frac{2}{\pi}}\left(\frac{e^{\varepsilon_2}-e^{-\varepsilon_2}}{2}\right)\left(\frac{e^{\varepsilon_1}-e^{-\varepsilon_1}}{2} \wedge 1\right)$ we are done else dividing both sides by $\sqrt{I_F(T;0)}$ we have

$$\sqrt{I_F(T;0)} \le n\sqrt{\frac{2}{\pi}}\left(\frac{e^{\varepsilon_2}-e^{-\varepsilon_2}}{2}\right)\left(\frac{e^{\varepsilon_1}-e^{-\varepsilon_1}}{2} \wedge 1\right)$$
$$+ \left(2n\delta_2 64\left(\log\frac{8}{\delta_2}\right)^2 + 2n\delta_2\right)n^{-1}\left(\frac{2}{\pi}\right)^{-1/2}\left(\frac{e^{\varepsilon_1}-e^{-\varepsilon_1}}{2} \wedge 1\right)^{-1}\left(\frac{e^{\varepsilon_2}-e^{-\varepsilon_2}}{2}\right)^{-1}$$

The second term can be dropped if $\delta_2 \log(1/\delta_2)^2 = o(n\varepsilon_1^2\varepsilon_2^2)$. The final form is achieved by using the fact that $\varepsilon_1, \varepsilon_2 \le 1$. □

### B.4 Auxiliary Lemmas

**Lemma 4.** *Define $G_k = \mathbb{E}(X_k \mid T_1)Y_k\left(\sum_{i=1}^n \mathbb{E}(\mathbb{E}(X_i \mid T_1)Y_i \mid T_1, T_2)\right)$ then we have that*

$$\int_M^\infty \mathbb{P}(|G_k| \ge t)dt \le 16(\sqrt{M})e^{-\sqrt{M}/4}$$

*Proof.* Let us denote by $Z_i = \mathbb{E}(X_i \mid T_1)Y_i$. We begin by bounding $\mathbb{E}e^{t|G_i|^{1/2}}$. By AM-GM and Cauchy-Schwarz inequality, we have that

$$\mathbb{E}e^{t|G_i|^{1/2}} = \mathbb{E}e^{t|Z_i|^{1/2}|\mathbb{E}(Z_i|T_1,T_2)|^{1/2}}$$
$$\le \mathbb{E}e^{\frac{1}{2}t(|Z_i|+|\mathbb{E}(Z_i|T_1,T_2)|)}$$
$$\le \sqrt{\mathbb{E}e^{t|Z_i|}}\sqrt{\mathbb{E}e^{t|\mathbb{E}(Z_i|T_1,T_2)|}}$$

Using the conditional Jensen's Inequality with the function $x \to e^{tx^2}$ which is convex to obtain that

$$\mathbb{E}e^{t|\mathbb{E}(Z_i|T_1,T_2)|} \le \mathbb{E}(\mathbb{E}(e^{t|Z_i|} \mid T_1, T_2)) = \mathbb{E}(e^{t|Z_i|})$$

Hence $\mathbb{E}e^{t|G_i|^{1/2}} \le \mathbb{E}(e^{t|Z_i|})$. Bounding the RHS as follows

$$\mathbb{E}(e^{t|Z_i|}) = \mathbb{E}e^{t|Y_i||\mathbb{E}(X_i|T_1)|}$$
$$\le \mathbb{E}e^{\frac{1}{2}t\left(Y_i^2 + (\mathbb{E}X_i|T_1)^2\right)}$$
$$\le \mathbb{E}e^{\frac{1}{2}tY_i^2}\mathbb{E}e^{\frac{1}{2}t(\mathbb{E}X_i|T_1)^2}$$

where we used the AM-GM inequality in the second line and the independence of $Y_i$ and $\mathbb{E}(X_i \mid T_1)$ in the third line. Using conditional Jensen again, we would have $\mathbb{E}e^{\frac{1}{2}t(\mathbb{E}X_i|T_1)^2} \le \mathbb{E}e^{\frac{1}{2}tX_i^2}$ which implies $\mathbb{E}(e^{t|Z_i|}) \le \mathbb{E}e^{\frac{1}{2}tX_i^2}\mathbb{E}e^{\frac{1}{2}tY_i^2}$.

Putting everything together we have that $\mathbb{E}e^{t|G_i|^{1/2}} \le \mathbb{E}e^{\frac{1}{2}tX_i^2}\mathbb{E}e^{\frac{1}{2}tY_i^2} \le 2$ for $t \le 1/2$ (since $X_i, Y_i \sim \chi_1^2$). This implies that

$$\mathbb{P}(|G_i| \ge t) \le \mathbb{P}(e^{\frac{1}{4}|G_i|^{1/2}} \ge e^{\sqrt{t}/4}) \le 2e^{-\sqrt{t}/4}.$$

The last inequality follows from Markov. Hence we have that

$$\int_M^\infty \mathbb{P}(|G_i| \ge t)\,dt \le 2\int_M^\infty e^{-\sqrt{t}/4}\,dt = 16(\sqrt{M})e^{-\sqrt{M}/4}.$$

□

**Lemma 5.** *Assuming for $k = 1, 2$, $\delta_k \log(1/\delta_k) = o(\varepsilon_k^2)$, we have for any $1 \leq i \leq n$*
$$\mathbb{E}(\mathbb{E}(X_i \mid T_1))^2 \leq \frac{2}{\pi} \left( \frac{e^{\varepsilon_1} - e^{-\varepsilon_1}}{2} \right)^2, \text{ similarly we have } \mathbb{E}(\mathbb{E}(Y_i \mid T_2))^2 \leq \frac{2}{\pi} \left( \frac{e^{\varepsilon_2} - e^{-\varepsilon_2}}{2} \right)^2.$$

*Proof of Lemma 5.* Note that $\mathbb{E}(\mathbb{E}(X_i \mid T_1))^2 = \mathbb{E}[X_i(\mathbb{E}(X_i \mid T_1))]$. Denote $A_i = X_i(\mathbb{E}(X_i \mid T_1))$ we can write $\mathbb{E}A_i = \mathbb{E}A_i^+ - \mathbb{E}A_i^-$. Also let us define $A_i' = X_i(\mathbb{E}(X_i \mid T_1'))$ where $T_1' = T_1(\boldsymbol{X}')$, where $\boldsymbol{X}'$ is the adjacent dataset with its $i$th data point replaced by $X_i'$ which is an independent copy.

We can write $\mathbb{E}A_i^+$ as

$$
\begin{aligned}
\mathbb{E}(A_i^+) &= \int_0^\infty \mathbb{P}(A_i^+ \geq t) dt \\
&= \int_0^M \mathbb{P}(A_i^+ \geq t) dt + \int_M^\infty \mathbb{P}(A_i^+ \geq t) dt \\
&\leq \int_0^M e^{\varepsilon_1} \mathbb{P}((A_i')^+ \geq t) dt + \delta_1 M + \int_M^\infty \mathbb{P}(A_i^+ \geq t) dt \\
&= e^{\varepsilon_1} \mathbb{E}(A_i')^+ - e^{\varepsilon_1} \int_M^\infty \mathbb{P}((A_i')^+ \geq t) dt + \delta_1 M + \int_M^\infty \mathbb{P}(A_i^+ \geq t) dt \\
&\leq e^{\varepsilon_1} \mathbb{E}(A_i')^+ + \delta_1 M + \int_M^\infty \mathbb{P}(|A_i| \geq t) dt
\end{aligned}
$$

Similarly we have that

$$
\begin{aligned}
\mathbb{E}(A_i^-) &= \int_0^\infty \mathbb{P}(A_i^- \geq t) dt \\
&= \int_0^M \mathbb{P}(A_i^- \geq t) dt + \int_M^\infty \mathbb{P}(A_i^- \geq t) dt \\
&\geq \int_0^M e^{-\varepsilon_1} \mathbb{P}((A_i')^- \geq t) dt - \delta_1 M + \int_M^\infty \mathbb{P}(A_i^- \geq t) dt \\
&= e^{-\varepsilon_1} \mathbb{E}(A_i')^- - e^{-\varepsilon_1} \int_M^\infty \mathbb{P}((A_i')^- \geq t) dt - \delta_1 M + \int_M^\infty \mathbb{P}(A_i^- \geq t) dt \\
&\geq e^{-\varepsilon_1} \mathbb{E}(A_i')^- - \int_M^\infty \mathbb{P}(|A_i'| \geq t) dt - \delta_1 M
\end{aligned}
$$

Since $\mathbb{E}A_i = \mathbb{E}A_i^+ - \mathbb{E}A_i^-$ we have that

$$
\begin{aligned}
\mathbb{E}A_i &\leq e^{\varepsilon_1} \mathbb{E}(A_i')^+ - e^{-\varepsilon_1} \mathbb{E}(A_i')^- + 2\delta_1 M + \int_M^\infty \mathbb{P}(|A_i| \geq t) dt + \int_M^\infty \mathbb{P}(|A_i'| \geq t) dt \\
&= \left( \frac{e^{\varepsilon_1} + e^{-\varepsilon_1}}{2} \right) \mathbb{E}A_i' + \left( \frac{e^{\varepsilon_1} - e^{-\varepsilon_1}}{2} \right) \mathbb{E}|A_i'| + 2\delta_1 M + \int_M^\infty \mathbb{P}(|A_i| \geq t) dt + \int_M^\infty \mathbb{P}(|A_i'| \geq t) dt \\
&= \left( \frac{e^{\varepsilon_1} - e^{-\varepsilon_1}}{2} \right) \mathbb{E}|A_i'| + 2\delta_1 M + \int_M^\infty \mathbb{P}(|A_i| \geq t) dt + \int_M^\infty \mathbb{P}(|A_i'| \geq t) dt \qquad (46)
\end{aligned}
$$

where we have used the fact that $\mathbb{E}A_i' = 0$. Observe that

$$\mathbb{E}|A_i'| = \mathbb{E}|X_i|\mathbb{E}|\mathbb{E}(X_i \mid T_1')| \leq \sqrt{\frac{2}{\pi}} \sqrt{\mathbb{E}(\mathbb{E}(X_i \mid T_1'))^2} = \sqrt{\frac{2}{\pi}} \sqrt{\mathbb{E}A_i}$$

Next we upper bound $\int_M^\infty \mathbb{P}(|A_i| \geq t) dt$ in that direction we look at

$$
\begin{aligned}
\mathbb{E}e^{t|A_i|} &= \mathbb{E}e^{t|X_i||\mathbb{E}(X_i|T_1)|} \\
&\leq \mathbb{E}e^{\frac{1}{2}t(X_i^2 + (\mathbb{E}(X_i|T_1))^2)}
\end{aligned}
$$

where we used the AM-GM inequality for the exponent. Next we can apply the Cauchy-Schwarz inequality to obtain that

$$\mathbb{E}e^{t|A_i|} \leq \sqrt{\mathbb{E}e^{tX_i^2}} \sqrt{\mathbb{E}e^{t\mathbb{E}(X_i|T_1))^2}}$$

the second term can further be bounded using the conditional Jensen's Inequality with the function $x \to e^{tx^2}$ which is convex to obtain that

$$\mathbb{E}e^{t\mathbb{E}(X_i|T_1))^2} \leq \mathbb{E}(\mathbb{E}(e^{tX_i^2} \mid T_1)) = \mathbb{E}(e^{tX_i^2})$$

Putting everything together we have that $\mathbb{E}e^{t|A_i|} \leq \mathbb{E}(e^{tX_i^2}) \leq \sqrt{2}$ for $t \leq 1/4$ (since $X_i \sim \chi_1^2$).This implies that

$$\mathbb{P}(|A_i| \geq t) \leq \mathbb{P}(e^{\frac{1}{4}|A_i|} \geq e^{t/4}) \leq \sqrt{2}e^{-t/4}.$$

The last inequality follows from Markov. Hence we have that $\int_M^\infty \mathbb{P}(|A_i| \geq t) \leq 4\sqrt{2}e^{-M/4}$, set $M = 4\log(1/\delta_1)$ to obtain $\int_M^\infty \mathbb{P}(|A_i| \geq t) \leq 4\sqrt{2}\delta_1$. we can similarly show that

$$\int_M^\infty \mathbb{P}(|A_i'| \geq t)dt \leq 4\sqrt{2}\delta_1.$$

Putting everything together we have that

$$\mathbb{E}A_i \leq \left(\frac{e^{\varepsilon_1} - e^{-\varepsilon_1}}{2}\right) \sqrt{\frac{2}{\pi}} \sqrt{\mathbb{E}A_i} + 8\delta_1 \log(1/\delta_1) + 8\sqrt{2}\delta_1$$

If $\mathbb{E}A_i \leq \frac{2}{\pi} \left(\frac{e^{\varepsilon_1} - e^{-\varepsilon_1}}{2}\right)^2$ we are done else dividing both sides by $\sqrt{\mathbb{E}A_i}$ we have

$$\sqrt{\mathbb{E}A_i} \leq \left(\frac{e^{\varepsilon_1} - e^{-\varepsilon_1}}{2}\right) \sqrt{\frac{2}{\pi}} + (8\delta_1 \log(1/\delta_1) + 8\sqrt{2}\delta_1) \left(\frac{2}{\pi}\right)^{-1/2} \left(\frac{e^{\varepsilon_1} - e^{-\varepsilon_1}}{2}\right)^{-1}$$

The second term can be dropped if $\delta_1 \log(1/\delta_1) = o(\varepsilon_1^2)$. $\qquad\square$

**Lemma 6.** *For square matrices $A$ and $B$, if $B$ is symmetric, we have*

$$\mathrm{tr}(AB) \leq \|A\|_2 \mathrm{tr}(B)$$

*Proof of Lemma 6.* The proof follows from von Neumann's trace inequality:

$$\mathrm{tr}(AB) \leq |\mathrm{tr}(AB)| \leq \sum_i \alpha_i \beta_i \leq \max(\alpha_i) \sum_i \beta_i = \max(\alpha_i) \times \mathrm{tr}(B)$$

where $\alpha_i$ and $\beta_i$ are the singular values of $A$ and $B$ respectively. The proof follows by the definition of $\ell_2$ operator norm used on matrix $A$. $\qquad\square$

