# OpenReview forum: "⁠When Data Can't Meet: Estimating Correlation Across Privacy Barriers"
_NeurIPS.cc/2025/Conference — NeurIPS 2025 spotlight_

### Official Review · Reviewer_TVPc · 2025-06-10

**Clarity:** 4
**Significance:** 4
**Originality:** 3
**Rating:** 5
**Confidence:** 4

**Summary:**

This paper proposes a series of novel solutions for privately estimating the correlation (equiv. covariance) of two random variables X and Y vertically partitioned to two servers. Its non-interactive protocol for Gaussian (NI, G) takes the sign of each value and releases batch means of each server with Lapalce noise. The covariance is then estimated though a dot product between the noisy means. For subgaussians (NI, SG), truncation is applied to achieve a similar bound with O(log(n)) overhead. In its interactive protocol (INT, G), the server with less stringent privacy requirement applies random response to (the sign of) each value and sends it to the other server, who then computes a similar privatized dot product for estimating the covariance. The interactive protocol for subgaussian (INT, SG) also involves truncation. The paper provided confidence intervals for the above methods and proved matching lower bounds. It also conducted experiments on simulated data to justify the performance.

**Questions:**

In addition to the weaknesses above, I have the following questions:

Line 93: In the second inequality, why is P(T2) conditioned on X? As T2 does not obvserve X directly, should the probability be conditioned on the same value of T1(X)?

Line 106: If X is public, then why consider constant eps1 instead of eps=infty? In the latter case, your error bound for the interactive protocal becomes zero, which does not make sense since Y is still private. According to the proof, the dependency on eps1 should be min{eps1, 1}, I think it is better to clarify this.

**Ethical Concerns:**

["NO or VERY MINOR ethics concerns only"]

**Final Justification:**

This paper proposes a series of novel solutions for privately estimating the correlation (equiv. covariance) of two random variables X and Y in the two server model. Although there are some limitations (e.g. supporting only two-servers, and only sub-gaussian distributions), the paper is in general technically dense, well-organized and clearly presented, provided that the typos are fixed as mentioned in the rebuttal. Therefore I would like to recommend accept.

**Limitations:**

Yes

**Quality:**

3

**Strengths And Weaknesses:**

S1. This paper is technically dense. It achieves matching upper and lower bounds for the proposed problem, which is significant.

S2. There are novel ideas in constructing the upper and lower bounds.

S3. The paper is well-organized with precise statements and logical proofs.

W1: For experiments, this paper provides simulation results where data are drawn from specific distributions. It will be interesting to perform experiments on real-world data which are not exactly Gaussian (or sub-Gaussian). In that case, the DP guarantee should still hold while the utility guarantee may not. But analyzing the estimation error still makes sense. Since the protocols require \eta_l as inputs, it also helps to discuss about how they can be estimated for real data.

W2: In Section 3 the authors provide an interactive protocol which achieves better rate than the non-interactive one. However, the main content of the paper do not discuss the reason behind this improvement. To be specific, the interactive protocol uses a different randomization method for T_1(X) compared to the non-interactive counterpart, it will be better to explain the intuition behind this design. Furthermore, the results in this paper holds for one-way interaction, it will be interesting to add a discussion on the two-way interaction results by properly dividing the privacy budget.

W3: There are several typos in the paper, which should be corrected:
Line 49: (eps1, delta) > (eps1, delta1)
The equation following line 64: T1(X)\in A | Y > T1(X)\in A | X
Theorem 1.1: Please define \asymp, which was used to denote big-Theta. Also define \lesssim and \gtrsim.
Line 74: is this case > in this case
Theorem 1.2: Please mention that \wedge (and \vee) was used for min (and max).
Line 136: The current definition of Bj looks like a set of 2 elements, please use {m(j-1)+1,...,mj}
Line 164: The noise should be 2*lambda_l / (m eps_l) here. This should be a typo, since the proof in line 760 looks fine.
Line 753, Equation (16): The presence of constants here indicates the analysis is not asymptotic. Then the first equality (3rd line) is incorrect. By symmetry, you have E[(X+Z)^4]=E[X^4] + 6E[X^2Z^2] + E[Z^4] and the equality does not make sense. I believe this is asymptotically correct, but please be careful about the constants since it affects your choice of m.
Line 757: The first equality needs to be corrected. You ommitted terms for X>lambda1 and Y>lambda2. Please include the condition for this and note that you should not write equal.
Line 760: Missing end bracket.

---

> ### Author Rebuttal · Authors · 2025-07-30
>
> First of all, thank you very much for the  for the favorable review  and the very careful reading. Your comments have helped us pinpoint a lot of oversights, which we have already corrected, and look forward to sharing in the camera ready version.
>
> ### Answer to Weakness 1
>
> In the revision we will add a real data experiment with the specific goal of explaining the use-case of our results. As you have pointed out, estimating the sub-Gaussianity parameter $\eta$ is a valid concern in real data applications. We have not focused on this issue in the main results, but will make this explicit in the experiments section in our camera ready version.
>
> While working under the sub-Gaussianity assumption, to use our estimators and confidence intervals, we suggest obtaining an upper bound for $\eta_1$ using the range of a subset of $X$, and part of the privacy budget $\varepsilon_1$. This task would incur a cost according to central privacy rates, which would be asymptotically negligible in comparison to all the minimax rates we have found for vertical privacy.
>
> ### Answer to Weakness 2
>
> The basic intuition is from the standpoint of whether data on the two servers can meet. In the strictest notion where data cannot meet, we find the non-interactive rates. _Relaxing_ this to allow one server to look at certain private transcripts from the other server can thus be expected to improve the estimation rates. The exact nature of the transcripts was derived after experimenting with how the size of the transcripts to be shared affects the final rate. The specific version of randomization was chosen using existing local-DP techniques.
>
> On two-way interactions, we conjecture that the rates cannot be improved beyond the currently derived ones. However our current framework does not cover two-way interactions, and so far such a general proof eludes us. We intend to pursue this in future research.
>
> ### Answer to Weakness 3
>
> Thank you for pointing out these typos! We will add a section explaining notations, as you and another reviewer have suggested. In addition, your comments on lines 753 and 757 would be answered as follows.
>
> Line 753: We will update equation 16 to $\left(
> 			\frac{9m}{n}
> 			+
> 			\frac{24}{n\varepsilon_1^2}
> 			+
> 			\frac{24}{n\varepsilon_1^2}
> 			+
> 			\frac{64}{mn\varepsilon_1^2\varepsilon_2^2}
> 			\right)	$, and replace the equality in the previous line with inequality, as you have correctly pointed out. Without changing the choice of $m$, this leads to the final upper bound of $\frac{10\pi^2}{n}\left(\frac{1}{\varepsilon_1}+\frac{1}{\varepsilon_2}\right)^2$, validating the current statement of our theorem. The constants can be made tighter for a different choice of $m$, again, following your suggestion. We will implement this in the camera ready version.
>
> Line 757: Indeed, there would be additional terms for when $|X|$ or $|Y|$ are larger than $\lambda_1$ or respectively $\lambda_2$. To deal with this, we will write out the four terms of the expectation explicitly --- each term being contributed for the four cases depending on whether none, exactly one, or both, of $|X|$ is larger, or $|Y|$ is larger.
>
> ### Answer to Question 1
>
> Yes, we will update line 93 to allow conditioning on T1, and not on X. That is, the updated display would read
> $$
> \mathbb{P}(T_2(\boldsymbol{Y},T_1)\in A|T_1,\boldsymbol{Y})\le~ \exp(\varepsilon_2)\mathbb{P}(T_2(\boldsymbol{Y}',T_1)\in A|T_1,\boldsymbol{Y}')+\delta_2.
> $$
>
> ### Answer to Question 2
>
> Thank you for pointing this out. In the revision, we will add $\varepsilon_1\wedge 1$ in place of $\varepsilon_1$ as you have suggested.

---

> > ### Comment · Reviewer_TVPc · 2025-08-02
> >
> > Thanks for the clarification, I have no further questions.

---

### Official Review · Reviewer_cn9Y · 2025-07-02

**Clarity:** 4
**Significance:** 3
**Originality:** 4
**Rating:** 5
**Confidence:** 2

**Summary:**

This paper studies the fundamental problem of estimating the Pearson correlation coefficient between two variables that are vertically separated across two servers. The authors establish the minimax optimal estimation error rates under server-level differential privacy. The analysis covers two distinct communication protocols: a non-interactive setting where servers release privatized statistics independently, and an interactive setting where one server can access the privatized output of the other. The core theoretical contribution is the derivation of matching upper and lower bounds for the estimation error, which reveals that the interactive protocol is provably superior if and only if the servers have asymmetric privacy budgets. The authors propose estimators that achieve these optimal rates and provide numerical experiments to support their theory.

**Questions:**

- How would your estimators need to be adapted for heavy-tailed distributions? Would techniques based on median-of-means or other robust statistics fundamentally alter the minimax rates you have established?

- Could you provide more insight into the non-asymptotic performance? A key concern with DP is that the hidden constants in the error bounds can render a method impractical. Can you characterize how the constants in your rates depend on the privacy parameters ($\epsilon$, $\delta$) and the sub-Gaussian parameters of the data? This is essential for understanding the true sample complexity of your method.

**Ethical Concerns:**

["NO or VERY MINOR ethics concerns only"]

**Final Justification:**

The authors have clearly addressed all my concerns. I believe this is a good paper, and my former score already adequately reflects my assessment.

**Limitations:**

The authors have adequately addressed the limitations of their work

**Paper Formatting Concerns:**

I noticed some potential deviations from the official NeurIPS 2025 style file, e.g.:

- The main text font appears slightly different from the standard Times New Roman used in the template;
- The bibliography's formatting seems to diverge from the standard style recommended by the official style file.

However, these are minor points that do not affect my assessment of the scientific content

**Quality:**

3

**Strengths And Weaknesses:**

### Strengths
1. The paper's primary strength is its rigorous comparison of non-interactive and interactive protocols, yielding the new insight that interaction's benefit is tied directly to asymmetric privacy budgets.

1. Within its assumed setting, the work is technically complete, providing matching upper and lower bounds on the minimax risk.

---

### Weaknesses
1. The paper's most significant weakness is its focus on the bivariate case. By assuming each server holds only a single data column, it solves a "toy problem" that bears little resemblance to real-world Vertical Federated Learning, where entities hold datasets with hundreds or thousands of features. This makes it difficult to see the derived rates as a meaningful benchmark for any practical VFL application.

1.  The analysis relies on the restrictive assumption of sub-Gaussian data, which excludes many important real-world datasets that exhibit heavy tails. This limits the applicability of the proposed estimators and the relevance of the optimality guarantees.

1. The paper's asymptotic analysis of the error rates hides the constant factors, which can be substantial and dictate the practical sample complexity. There is a lack of discussion on the number of samples $n$ required to achieve reasonable error under strong privacy settings (e.g., small $\epsilon$), which is a crucial question for any real-world deployment.

---

> ### Author Rebuttal · Authors · 2025-07-30
>
> Thank you for the favorable review and the very pertinent questions. Please find below our responses to each of your comments.
>
> ### Answer to Weakness 1
> A very valid concern indeed. In the multidimensional setting we are estimating a correlation matrix, and we expect the rates to depend on 1) dimension $d$, and 2) the norm which we use to estimate the matrix. We plan to address this point in the future work section.
> ### Answer to Weakness 2 and Question 1
> The heavy-tailed case is fundamentally different from the sub-Gaussian case, even in the case of mean estimation and central DP. For example, KSU20 consider mean estimation under central DP through a median of means, and show a worse minimax rate due to the heavy tails. We expect this fundamental difference (in terms of rate) to also prevail in this vertical FL setup and consider it as an avenue for future work.
>
> KSU20: Kamath, G., Singhal, V., Ullman, J.. (2020). Private Mean Estimation of Heavy-Tailed Distributions. <i>Proceedings of Thirty Third Conference on Learning Theory</i>, in <i>Proceedings of Machine Learning Research</i> 125:2204-2235.
>
>
>
> ### Answer to Weakness 3 and Question 2
> The rates in the current version are written to explicitly show the dependence on the privacy parameters $\varepsilon_1$ and $\varepsilon_2$. Following your suggestion, in the revision, we will also make the dependence on the subgaussian parameters ($\eta_1$ and $\eta_2$) more explicit. Upto numerical constants, the sub-Gaussian error rates in the non-interactive case would be $\frac{\log(n)}{n}
> 		\left(
> 		\frac{\eta_1^2}{\varepsilon_1^2}
> 		+\frac{\eta_2^2}{\varepsilon_2^2}
> 		\right)$. For the interactive case, the sub-Gaussian error rates would be $\frac{\log(n)}{n(\frac{\varepsilon_1}{\eta_1}\vee \frac{\varepsilon_2}{\eta_2})^2}
> 	+\frac{\eta_1^2\vee\eta_2^2(\log(n))^4}{n^2\varepsilon_1^2\varepsilon_2^2}$.
> All of this would be made more precise in the camera ready version.

---

> > ### Comment · Reviewer_cn9Y · 2025-08-03
> >
> > The answers have addressed all my concerns. Thanks.

---

### Official Review · Reviewer_FZy2 · 2025-07-08

**Clarity:** 3
**Significance:** 2
**Originality:** 3
**Rating:** 5
**Confidence:** 4

**Summary:**

This work focuses on federated learning for correlation estimation of two random variables vertically partitioned on two servers. Due to privacy issues, servers share only privatized data with two sets of privacy budgets in terms of differential privacy. This work utilizes minimax optimal rates for correlation estimation for both non-interactive and interactive mechanisms, and also develops confidence intervals for the correlations. Additionally, it proves that interactive mechanism is better than the non-interactive one.

**Questions:**

1. Why does the delta on both servers does not affect the minimax rate?
2. Why does the conditional probability in the right-hand side of the second inequality under Line 93 depend on X,Y’, not X’, Y’? In other word, why does the T2 construction not depend on X’?
3. Line 99, I think you want to say interactive instead of non interactive.
4. What datasets do you use in the experiments? Would be better to use some well-known benchmarks.
5. What will be different if this work extends to several features instead of one feature? Can you talk more on this for the future work? It is ok to include this point in the appendix if space is limited.

**Ethical Concerns:**

["NO or VERY MINOR ethics concerns only"]

**Final Justification:**

Authors have cleared all my questions, and since all reviewers tend to accept it, I will keep my score, too.

**Quality:**

3

**Strengths And Weaknesses:**

Strengths:

1. This work provide both interactive and non-interactive framework for the correlation estimation with theorems, and verify that the interactive one outperforms the non-interactive one.
2. This work considers the privacy budgets on both servers instead of a central privacy budget.
3. Except theorems, this work provides experiments for both cases, and open-sourced the code.

Weakness:

1. The authors did not mention why they focus on correlation estimation, and why this problem is important in the VFL with DP.
2. This work  hides all information of proofs in the appendixe. Would be better to mention clues to prove each theorem in the main text with one or two sentences.

---

> ### Author Rebuttal · Authors · 2025-07-30
>
> Thank you for the favorable review and the very relevant questions. Please find below our responses to the issues you have pointed out.
>
> ### Response to Weakness 1
> Thank you for this question on the motivation of the current work. We decided to explore and demonstrate the cost of privacy for correlation estimation because correlation is the most canonical example of interest from a _joint_ distribution. Just as the mean is arguably the most basic object to understand in the setting of central differential privacy, correlation becomes the essential quantity of interest in a joint distribution, even more so, when two sides of the data cannot meet. As our results show, this natural object already shows interesting departures from traditional rates on differential privacy. We believe exploring vertical federated learning through the lens of privacy cost would lead to even more exciting results--- our exploration into the correlation estimation is merely the tip of the iceberg. A number of interesting future directions have already appeared, from your and the other reviewers' suggestions (e.g., in your Question 4), which we will explore in subsequent work.
>
> We appreciate your suggestion on including a short description of the proof. Space permitting, we will include steps for the upper bounds in the camera ready version. Note that the proof strategy for the lower bound is already given in Section 4.
>
>
> ### Response to Question 1
> For the one-dimensional problem, both pure DP $(\varepsilon,0)$ and approximate DP $(\varepsilon,\delta)$ achieve the same rate—provided $\delta$ is sufficiently small. In other words, relaxing from pure to approximate DP does not improve the information‑theoretic rate.
>
> ### Response to Question 2
> Thanks for the great question; equation 9 is a key point in our privacy definition. To avoid confusion we will update the second line to:
> $$
> \mathbb{P}(T_2(\boldsymbol{Y},T_1)\in A|T_1,\boldsymbol{Y})\le~ \exp(\varepsilon_2)\mathbb{P}(T_2(\boldsymbol{Y}',T_1)\in A|T_1,\boldsymbol{Y}')+\delta_2
> $$
> with the interpretation that in the interactive case, we are augmenting the non-interactive definition to allow $T_2$ to depend on $T_1$, which is seen as a fixed output from server 1. Since $T_1$ is already private, this inequality pertains to the privacy requirements on $\boldsymbol{Y}$ for a fixed $T_1$. See also the first question by Reviewer TVPc.
>
> ### Response to Question 3
>
> Yes, that is correct; we shall correct this typo in the camera-ready version.
>
> ### Response to Question 4
>
> Our numerical study sections are based on synthetic data (generated via simulations). We plan to add real data examples in the camera-ready version.
>
> ### Response to Question 5
>
> In the multidimensional setting one would have to estimate a correlation matrix, and we expect the rates to depend on 1) dimension $d$, and 2) the norm which we use to estimate the matrix. We plan to address this point in the future work section.

---

> > ### Comment · Reviewer_FZy2 · 2025-08-06
> >
> > Thanks for your reply! You claried all questions and including real dataset in the empirical results is a good idea.

---

### Official Review · Reviewer_dVhn · 2025-07-23

**Clarity:** 2
**Significance:** 4
**Originality:** 4
**Rating:** 5
**Confidence:** 3

**Summary:**

This paper considers the problem of estimating the correlation between two random variables, $X$ and $Y$, which are obfuscated by approximate (i.e., $(\epsilon,\delta)$) differential privacy mechanisms. The paper considers two main scenarios:

- Non-interactive (NI) Protocol: In this setting, the statistics about $X$ and $Y$ are constructed and privatized independently.
- Interactive (INT) Protocol: In this setting, the statistic about one variable (say, $X$) is constructed and sanitized independently from $Y$, whereas the statistic about the other variable $Y$ is constructed with the benefit of accessing the privatized version of the statistics of $X$, and then privatized.

The paper derives approximate formulas for the minimax optimal rate for correlation estimation in these two scenarios for Gaussian and sub-Gaussian distributions, in terms of MSE. One of the main results is that (perhaps unsurprisingly) the estimation under INT is strictly more precise than under NI.

The paper also proposes methods to estimate the correlations in both scenarios, which also provide confidence intervals. It also proves minimax lower bounds, confirming that the proposed estimation methods are nearly optimal. Another result (which, on the contrary, I found quite surprising) is that the formula for the NI case does not depend on $\delta$.

Finally, the paper presents experimental results that validate the theoretical findings.

**Questions:**

- Is $\rho$ the Pearson correlation coefficient?

- On what distribution on $\rho$ is the expectation in the Theorems 1.1. and 1.2 taken?

**Ethical Concerns:**

["NO or VERY MINOR ethics concerns only"]

**Final Justification:**

The authors have replied to my questions in their rebuttal in a convincing way

**Limitations:**

- I would suggest the authors to provide some experiments on real data to explore the practical relevance of the results.

-  I would recommend the authors to explain all the non-standard symbols and the key notions before using them.

**Paper Formatting Concerns:**

I did not see any issue

**Quality:**

3

**Strengths And Weaknesses:**

## Strengths

- The paper addresses an important problem and it justifies it well in terms of federated learning, specifically, for the vertically distributed setting. For instance, the case of healthcare data sharing between a hospital and a pharmaceutical company, where direct exchange of original data is not possible due to privacy constraints.

- The paper provides a thorough theoretical analysis of the specific cases of Gaussian and sub-Gaussian, and seems rather solid technically.

- I found it very surprising that the formula for the NI case does not depend on $\delta$.

- To the best of my knowledge, the paper addresses an original problem (although very natural), and the contribution is novel.

## Weaknesses

- I found the paper rather hard to read. First of all,  some notations are not explained. For instance, the symbol between the LHS and the RHS of the equations in Theorem 1.1 and 1.2 (I suppose it means "approximately equal to"), and the symbol "V" in the formula in Theorem 1.2 (I suppose it means "the larger of the two"). Furthermore, some key notions are not explained. For instance, the correlation measure $\rho$ (I assume it is the Pearson correlation coefficient). These shortcomings, however, can be fixed by a brief preliminary section, which I invite the authors to provide.

- The study is purely theoretical and on ideal distributions; it would have been good to provide at least one experiment on real data to explore the practical relevance of the results.

---

> ### Author Rebuttal · Authors · 2025-07-30
>
> Thank you for the positive review and the very relevant questions! Please find below our responses to the issues you have mentioned.
>
> ### Response to Weakness 1  and Limitation 2
> Thank you for bringing up the notational aspect. We agree that a concise preliminary section on our notation would greatly aid the reader. In the camera‑ready version, we will include such a section to ensure all symbols and conventions are clearly defined.
>
> ### Response to Weakness 2  and Limitation 1
> We appreciate your suggestion regarding empirical validation on heavy‑tailed data. For the camera‑ready version, we will incorporate additional experiments on real-world, heavy‑tailed datasets to demonstrate the robustness of our method. In addition, our preliminary findings suggest that heavy tailed data incur a stricter cost of privacy. We will investigate this in future work, and mention this direction in the future work section of this paper.
>
> ### Response to Question 1
> Yes, $\rho$ is indeed the Pearson correlation coefficient.
>
> ### Response to Question 2
> Our focus is on the minimax risk. Accordingly, we do not treat $\rho$ as a random variable nor impose any prior distribution on it. Instead, we evaluate the mean squared error for each fixed $\rho\in[-1,1]$ and report the worst‑case (maximum) risk over that interval.

---

> > ### Comment · Reviewer_dVhn · 2025-08-01
> >
> > I would  like to thank the authors for the clarifications to my questions, and for their willingness to include a concise preliminary section to introduce the notation.

---

### Decision · Program_Chairs · 2025-09-17

**Decision:**

Accept (spotlight)

**Comment:**

The paper addresses the problem of estimating the correlation between two random variables whose values are stored on different servers and where differential privacy constraints are applied separately on each server. They derive the minimax optimal rates for both the interactive and non-interactive settings and algorithms that match these rates, showing that interactive estimators achieve strictly better rates.

Reviewers unanimously found the paper to be rigorous and substantial, tackling a novel but simple and well motivated problem and providing complete and technically sophisticated results. Congratulations on a very well received paper.

The weaknesses raised were minor in comparison, but included questions about application to real-world data and limitations to a relatively simple setting (bivariate, sub-Gaussian data), as well as some minor issues regarding notation and mathematical statements.

In the rebuttal, the authors responded to specific questions and promised to run experiments on one or more real-world data sets with heavy-tailed data for the final version of the paper. They highlighted that extensions to the multidimensional case and theoretical analysis for heavy-tailed distributions are interesting avenues for future work.